# Diffusion Counterfactual Generation with Semantic Abduction

**Rajat Rasal** [1]   **Avinash Kori** [1]   **Fabio De Sousa Ribeiro** [1]   **Tian Xia** [1]   **Ben Glocker** [1]

## Abstract

Counterfactual image generation presents significant challenges, including preserving identity, maintaining perceptual quality, and ensuring faithfulness to an underlying causal model. While existing auto-encoding frameworks admit semantic latent spaces which can be manipulated for causal control, they struggle with scalability and fidelity. Advancements in diffusion models present opportunities for improving counterfactual image editing, having demonstrated state-of-the-art visual quality, human-aligned perception and representation learning capabilities. Here, we present a suite of diffusion-based causal mechanisms, introducing the notions of spatial, semantic and dynamic abduction. We propose a general framework that integrates semantic representations into diffusion models through the lens of Pearlian causality to edit images via a counterfactual reasoning process. To our knowledge, this is the first work to consider high-level semantic identity preservation for diffusion counterfactuals and to demonstrate how semantic control enables principled trade-offs between faithful causal control and identity preservation.

## 1. Introduction

Answering counterfactual queries is central to understanding cause-and-effect relationships between variables in a system (Pearl, 2009; Peters et al., 2017; Bareinboim et al., 2022). Humans pose counterfactuals when reasoning creatively about *what could have been* under retrospective, hypothetical scenarios (Pearl, 2019). There has been increasing interest in counterfactual reasoning about images using deep generative models (Komanduri et al., 2023; Zečević et al., 2022). These advances are crucial for applications such as improving healthcare outcomes through personalised treatment (e.g., "what would a patient's tumour progression have looked like had they undergone a different treatment?") or enabling fairer AI by disentangling attributes in face modelling (e.g., "what would this person look like if their age, hairstyle or expression changed?"). Despite methodological advancements, high-fidelity counterfactual generation faces challenges with perceptual quality, identity preservation, and faithfulness to an underlying causal model (Monteiro et al., 2023; Melistas et al., 2024).

Diffusion models (Sohl-Dickstein et al., 2015; Song & Ermon, 2019; Ho et al., 2020) have achieved state-of-the-art performance in image synthesis (Dhariwal & Nichol, 2021; Podell et al., 2023) and exhibit strong alignment with human perception on recognition tasks (Jaini et al., 2023; Clark & Jaini, 2024), making them ideal candidates for improving automated creative reasoning through causal representation learning. Their popularity has grown in tasks related to counterfactual inference, such as image-to-image translation (Meng et al., 2021; Yang et al., 2023a) and image editing (Couairon et al., 2022; Hertz et al., 2022; Epstein et al., 2023), gradually supplanting traditional GAN and VAE-based approaches (Goodfellow et al., 2014; Karras, 2019; Karras et al., 2020; Kingma, 2013; Sohn et al., 2015). However, unlike closely related hierarchical generative models (Luo, 2022; Ribeiro & Glocker, 2024), where compact latent projections encode high-level semantics (Child, 2020; Vahdat & Kautz, 2020; Shu & Ermon, 2022), vanilla diffusion models lack a controllable semantic representation space (Turner et al., 2024). This limitation prohibits their applicability with existing counterfactual generation frameworks. Current methods introduce semantics into diffusion by interpreting low-rank projections of intermediate images (Park et al., 2023; Haas et al., 2024; Wang et al., 2024) or by employing diffusion models as decoders within VAEs (Preechakul et al., 2022; Pandey et al., 2022; Zhang et al., 2022; Batzolis et al., 2023; Abstreiter et al., 2021), where encoders capture controllable semantics. For explicit control, essential for counterfactual reasoning, diffusion models rely on guidance using discriminative score functions, either amortised within the diffusion model (Ho & Salimans, 2022; Karras et al., 2024; Chung et al., 2024; Dinh et al., 2023) or trained independently (Dhariwal & Nichol, 2021). However, challenges with identity preservation and faithful causal control — critical for image edits to align with our understanding of the world — still persist.

[1]Department of Computing, Imperial College London, UK. Correspondence to: Rajat Rasal <rajat.rasal17@imperial.ac.uk>.

*Proceedings of the 42$^{nd}$ International Conference on Machine Learning*, Vancouver, Canada. PMLR 267, 2025. Copyright 2025 by the author(s).

To address these challenges, the framework of structural causal models (SCMs) (Pearl, 2009; Peters et al., 2017; Bareinboim et al., 2022) has been extended with deep generative models, resulting in *deep* SCMs (DSCMs). These are used for structured counterfactual inference via a simple three-step procedure: *abduction-action-prediction* (Pawlowski et al., 2020; De Sousa Ribeiro et al., 2023; Xia et al., 2023; Wu et al., 2024; Dash et al., 2022; Sauer & Geiger, 2021). These methods typically parameterise their image-generating components using normalising flows (Papamakarios et al., 2021; Winkler et al., 2019), VAEs (Kingma, 2013; Sohn et al., 2015), GANs (Goodfellow et al., 2014; Mirza & Osindero, 2014), and HVAEs (Vahdat & Kautz, 2020; Child, 2020). More recently, diffusion models have been integrated into DSCMs to leverage their superior perceptual quality and explore counterfactual identifiability (Sanchez & Tsaftaris, 2022; Komanduri et al., 2024; Pan & Bareinboim, 2024). However, these efforts often overlook key nuances of guidance and representation learning unique to diffusion models, and their faithfulness to causal assumptions. This is particularly limiting for real-world applications, where backgrounds, illumination changes, and artefacts complicate causal modelling (Anonymous, 2024; Mokady et al., 2023; Lin et al., 2024). Additionally, SCMs have been informally applied to text-conditional diffusion editing (Song et al., 2024; Gu et al., 2023; Prabhu et al., 2023), yet text control alone often fails to provide the precision needed for real-world counterfactual reasoning.

We argue that diffusion models hold potential far beyond their established applications for recognition and image synthesis at the associative ($\mathcal{L}_1$) and interventional ($\mathcal{L}_2$) rungs on Pearl's ladder of causation (Pearl, 2009). Specifically, they can serve as foundations for counterfactual reasoning frameworks at $\mathcal{L}_3$, encompassing applications at $\mathcal{L}_1$ and $\mathcal{L}_2$, to enable principled decision-making. In this work, we take a significant step in this direction by revealing trade-offs inherent to using diffusion for image counterfactuals. We do this alongside integrating semantic control and high-level identity preservation into diffusion models via the DSCM framework. Our main contributions are as follows:

1. We present a causal generative modelling framework for counterfactual image generation using diffusion models (Section 3.1).

2. We introduce semantic abduction to improve counterfactual fidelity and identity preservation (Section 3.2).

3. We improve intervention-faithfulness using efficient amortised anti-causal guidance (Section 3.3).

4. We propose dynamic semantic abduction to infer semantics at each step in the diffusion process, better preserving backgrounds and specific characteristics in real-world images (Section 3.3).

## 2. Background

### 2.1. Causality

**Structural Causal Models.** We assume Markovian SCMs, referred to simply as SCMs henceforth, defined via the tuple $\mathfrak{S} := (F, \mathcal{E}, X)$ consisting of a set of functions $F = \{f_k\}_{k=1}^K$ called mechanisms, a set of observed variables $X = \{\mathbf{x}_k\}_{k=1}^K$, and a set of exogenous noise variables $\mathcal{E} = \{\boldsymbol{\epsilon}_k\}_{k=1}^K$ which are jointly independent, $p(\mathcal{E}) = \prod_{k=1}^K p(\boldsymbol{\epsilon}_k)$. Each mechanism models causal relationships as acyclic, invertible assignments of the form $\mathbf{x}_k := f_k(\mathbf{pa}_k, \boldsymbol{\epsilon}_k)$, where $\mathbf{pa}_k \subseteq X \backslash \{\mathbf{x}_k\}$ are $\mathbf{x}_k$'s direct causes or *parents* and $\boldsymbol{\epsilon}_k \sim p(\boldsymbol{\epsilon}_k)$. This induces a conditional factorisation of the observational joint distribution satisfying the causal Markov condition, $p_{\mathfrak{S}}(X) = \prod_{k=1}^K p_{\mathfrak{S}}(\mathbf{x}_k | \mathbf{pa}_k)$, whereby all observed variables are independent of their ancestors given their parents. An SCM can, therefore, be represented using a Bayesian network, known as a causal graph, where mechanisms are represented by edges going from observed ($\mathbf{pa}_k$) and exogenous causes ($\boldsymbol{\epsilon}_k$) to effect ($\mathbf{x}_k$), $\mathbf{pa}_k \rightarrow \mathbf{x}_k \leftarrow \boldsymbol{\epsilon}_k$.

**Counterfactuals.** SCMs can be used to answer retrospective, hypothetical questions, known as counterfactuals, of the form "Given observations $X$, what would $\mathbf{x}_k$ be had $\mathbf{pa}_k$ been different", via the following three-step procedure:

1. Abduction: Sample the exogenous posterior $\widetilde{\mathcal{E}} \sim p_{\mathfrak{S}}(\mathcal{E}|X) = \prod_{\mathbf{x}_k \in X} p_{\mathfrak{S}}(\boldsymbol{\epsilon}_k|\mathbf{x}_k, \mathbf{pa}_k)$ by inverting the parent mechanisms $\boldsymbol{\epsilon}_k = f_k^{-1}(\mathbf{x}_k, \mathbf{pa}_k)$.

2. Action: Perform an intervention on $\mathbf{x}_j \in \mathbf{pa}$ by fixing the output of $f_j(\cdot)$ to $b$, denoted $do(\mathbf{x}_j := b)$. The set of modified mechanisms is denoted as $\widetilde{F}$.

3. Prediction: Use $\widetilde{\mathfrak{S}} = (\widetilde{F}, \widetilde{\mathcal{E}}, X)$ to infer counterfactuals of interest using their mechanisms, $\widetilde{\mathbf{x}}_k = f_k(\boldsymbol{\epsilon}_k, \widetilde{\mathbf{pa}}_k)$, where $\widetilde{\mathbf{pa}}_k$ denotes the counterfactual parents under interventions.

Counterfactual inference adheres to the axiomatic properties of *composition*, *reversibility*, and *effectiveness* (Galles & Pearl, 1998; Halpern, 2000), framed as evaluation metrics by Monteiro et al. (2023), detailed in Appendix A.1. Composition measures $L_1$-reconstruction error by comparing observed images with counterfactuals under null interventions. Reversibility assesses cycle-consistency by measuring the $L_1$-distance between observed images and their cycled-back counterfactuals. Effectiveness evaluates faithfulness to interventions using anti-causal predictors for classification or regression. To measure a model's identity preservation (IDP), we compute LPIPS (Zhang et al., 2018) between compositions and their corresponding counterfactuals.

## 2.2. Diffusion Models

Denoising diffusion probabilistic models (DDPMs) are a class of latent variable model (Ho et al., 2020; Sohl-Dickstein et al., 2015; Song & Ermon, 2019) trained to progressively remove noise from samples $\mathbf{x}_T \sim \mathcal{N}(\mathbf{0}, \boldsymbol{I}) = p(\mathbf{x}_T)$ over $T$ timesteps to reveal an image $\mathbf{x}_0$. We focus on the deterministic variant of DDPMs, called denoising diffusion implicit models (DDIMs) (Song et al., 2020a). For conditional image generation, their generative model is

$$p_\theta(\mathbf{x}_{0:T}|\mathbf{c}) := p(\mathbf{x}_T) \prod_{t=1}^{T} p_\theta(\mathbf{x}_{t-1}|\mathbf{x}_t, \mathbf{c}),$$

$$p_\theta(\mathbf{x}_{t-1}|\mathbf{x}_t, \mathbf{c}) := q(\mathbf{x}_{t-1}|\mathbf{x}_t, d_\theta(\mathbf{x}_t, \mathbf{c}, t)), \quad (1)$$

the inference distribution uses linear-Gaussians transition,

$$q(\mathbf{x}_{1:T}|\mathbf{x}_0) := q(\mathbf{x}_T|\mathbf{x}_0) \prod_{t=2}^{T} q(\mathbf{x}_{t-1}|\mathbf{x}_t, \mathbf{x}_0)$$

$$q(\mathbf{x}_{t-1}|\mathbf{x}_t, \mathbf{x}_0) := \mathcal{N}(\boldsymbol{\mu}(\mathbf{x}_0, \mathbf{x}_t, t), \mathbf{0}), \quad (2)$$

$$q(\mathbf{x}_t|\mathbf{x}_0) := \mathcal{N}(\mathbf{x}_t; \sqrt{\alpha_t}\mathbf{x}_0, (1-\alpha_t)\boldsymbol{I}),$$

where $\boldsymbol{\mu}(\mathbf{x}_0, \mathbf{x}_t, t, t-1) =$

$$\sqrt{\alpha_{t-1}}\mathbf{x}_0 + \sqrt{1-\alpha_{t-1}}\frac{\mathbf{x}_t - \sqrt{\alpha_t}\mathbf{x}_0}{\sqrt{1-\alpha_t}}, \quad (3)$$

the denoiser $d_\theta(\cdot)$ is defined as

$$d_\theta(\mathbf{x}_t, \mathbf{c}, t) = \frac{1}{\sqrt{\alpha_t}}(\mathbf{x}_t - \sqrt{1-\alpha_t}\boldsymbol{\epsilon}_\theta(\mathbf{x}_t, \mathbf{c}, t)), \quad (4)$$

$\boldsymbol{\epsilon}_\theta(\cdot)$ is the noise estimator parametrised by a UNet (Ronneberger et al., 2015) and $\alpha_{0:T} \in [0, 1]$ defines the noise schedule. The image marginal $p_\theta(\mathbf{x}_0|\mathbf{c})$ is learned by optimising an evidence-lower bound via a surrogate objective: sample a timestep $t \sim \mathcal{U}[1, T]$, create noisy images by reparametrising $q(\mathbf{x}_t|\mathbf{x}_0)$ as $\hat{\mathbf{x}}_t = \sqrt{\alpha_t}\mathbf{x}_0 + \sqrt{1-\alpha_t}\boldsymbol{\epsilon}$ with $\boldsymbol{\epsilon} \sim \mathcal{N}(\mathbf{0}, \boldsymbol{I})$, then use $\boldsymbol{\epsilon}_\theta(\cdot)$ to predict $\boldsymbol{\epsilon}$:

$$\operatorname*{argmin}_\theta \{E_{\mathbf{x}_0, \mathbf{c}, t, \boldsymbol{\epsilon}}\|\boldsymbol{\epsilon} - \boldsymbol{\epsilon}_\theta(\hat{\mathbf{x}}_t, \mathbf{c}, t)\|_2^2\}. \quad (5)$$

We denote generative transitions as

$$\mathbf{x}_{t-1} = h_{t|\mathfrak{C}}(\mathbf{x}_t) = \boldsymbol{\mu}(d_\theta(\mathbf{x}_t, \mathbf{c}, t), \mathbf{x}_t, t, t-1), \quad (6)$$

which can also be inverted as

$$\mathbf{x}_t = h_{t|\mathfrak{C}}^{-1}(\mathbf{x}_{t-1})$$

$$= \boldsymbol{\mu}(d_\theta(\mathbf{x}_{t-1}, \mathbf{c}, t-1), \mathbf{x}_{t-1}, t-1, t) \quad (7)$$

where we use $\mathfrak{C} = \{\mathbf{c}, \theta\}$ to denote the *partial application* set which we abuse to include model parameters.

## 3. Methods

This section introduces our diffusion-based mechanisms, each distinguished by their exogenous noise decomposition and associated abduction. For simplicity, we denote an image in our SCM as $\mathbf{x}$, its parents as $\mathbf{pa}$, and its exogenous noise as $\boldsymbol{\epsilon}$. The image-generating mechanism $f(\cdot)$ is parameterised by a diffusion model with parameters $\theta$: $\mathbf{x} := f_\theta(\boldsymbol{\epsilon}, \mathbf{pa})$. The counterfactual conditions, under interventions, are denoted $\widetilde{\mathbf{pa}}$ and used to generate an image counterfactuals $\widetilde{\mathbf{x}}$ as: $\widetilde{\mathbf{x}} := f_\theta(\boldsymbol{\epsilon}, \widetilde{\mathbf{pa}})$, where $\boldsymbol{\epsilon} := f_\theta^{-1}(\mathbf{x}, \mathbf{pa})$. We provide definitions for a variety of abduction procedures which implement $\boldsymbol{\epsilon} := f_\theta^{-1}(\mathbf{x}, \mathbf{pa})$.

### 3.1. Spatial Mechanism

Our spatial mechanism is defined using the DDIM generative transitions in Equation (6):

$$\mathbf{x} := f_\theta(\boldsymbol{\epsilon}, \mathbf{pa}) \approx h_\theta(\mathbf{u}, \mathbf{pa})$$

$$:= (h_{1|\mathfrak{C}} \circ \cdots \circ h_{T|\mathfrak{C}})(\mathbf{u}), \quad (8)$$

where $\mathfrak{C} = \{\mathbf{pa}, \theta\}$, the observed variable $\mathbf{x}$ in our SCM is defined to be the DDIM generated image $\mathbf{x}_0$, $h_\theta(\cdot)$ samples the image marginal $p_\theta(\mathbf{x}|\mathbf{pa})$ and $p(\mathbf{u}) = p(\mathbf{x}_T)$ is the spatial exogenous prior. *Spatial abduction* is performed by sampling the spatial exogenous posterior:

$$p_{\mathfrak{S}}(\mathbf{u}|\mathbf{x}, \mathbf{pa}) \approx \delta(\mathbf{u} - h_\theta^{-1}(\mathbf{x}, \mathbf{pa})), \quad (9)$$

where $\delta(\cdot)$ denotes a Dirac delta distribution $h_\theta^{-1}(\cdot)$ is implemented using inverse DDIM transitions (Equation (7)) such that:

$$\mathbf{u} := f_\theta^{-1}(\mathbf{x}, \mathbf{pa}) \approx h_\theta^{-1}(\mathbf{x}, \mathbf{pa})$$

$$:= (h_{T|\mathfrak{C}}^{-1} \circ \cdots \circ h_{1|\mathfrak{C}}^{-1})(\mathbf{x}). \quad (10)$$

Here, the abducted $\mathbf{u}$ is a noisy and highly-editable version of $\mathbf{x}$ (Mokady et al., 2023; Hertz et al., 2022; Parmar et al., 2023) which encodes low-level structural information in the image space (Wang & Vastola, 2023). This differs from existing VAE and HVAE-based mechanisms (Pawlowski et al., 2020; De Sousa Ribeiro et al., 2023), which infer spatial noise through linear reparamerisation for intensity-/contrast correction, whereas our formulation allows more flexibility by inferring complex structural information via DDIM inversion. After performing interventions, we infer counterfactual conditions $\widetilde{\mathbf{pa}}$, then seed the counterfactual prediction step with $\mathbf{u}$ to generate image counterfactuals as

$$\widetilde{\mathbf{x}} := f_\theta(\boldsymbol{\epsilon}, \widetilde{\mathbf{pa}}) \approx h_\theta(\mathbf{u}, \widetilde{\mathbf{pa}}), \quad (11)$$

depicted via the twin network representation in Figure 1a. Our generalised spatial mechanism enables modelling multiple discrete or continuous parents of $\mathbf{x}$, thereby improving upon the DiffSCM framework (Sanchez & Tsaftaris, 2022), which only allows for a single discrete parent.

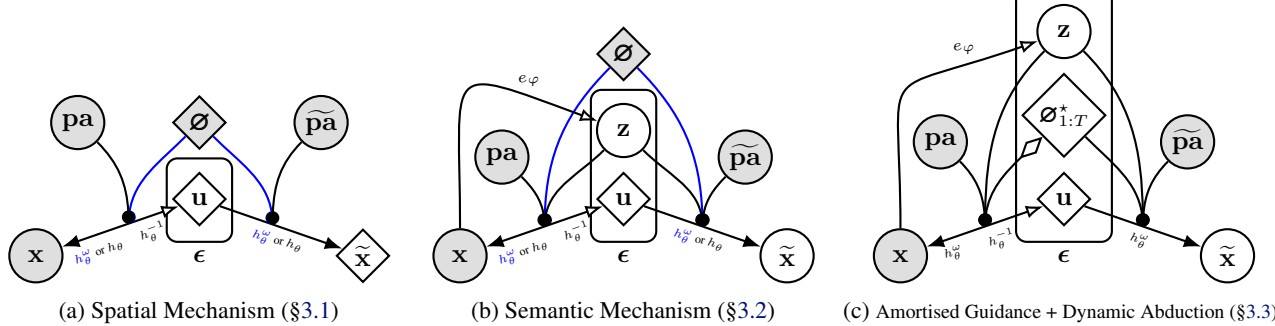

(a) Spatial Mechanism (§3.1)     (b) Semantic Mechanism (§3.2)     (c) Amortised Guidance + Dynamic Abduction (§3.3)

*Figure 1.* Twin network representations for our diffusion mechanisms. Black and white arrowheads refer resp. to the generative and abductive / inference directions. Edges ending in black circles depict conditions. Circular and diamond nodes refer resp. to depict random and deterministic variables. Boxes house the independent exogenous decomposition of $\boldsymbol{\epsilon}$. In (a)-(b), generation and inference are performed resp. with DDIM $h_\theta(\cdot)$ and DDIM inversion $h_\theta^{-1}(\cdot)$. Optionally, amortised guidance can be used by incorporating blue functions and conditions; $h_\theta^\omega(\cdot)$ for generation and $\varnothing$ for conditioning. In (b)-(c), the probabilistic encoder is depicted via $e_\varphi(\cdot)$. In (c), amortised guidance for the semantic mechanism is compulsory, and the diamond arrowhead denotes dynamic abduction.

Recall that counterfactual functions should be sound in terms of their *composition* (reconstruction) and *reversibility* (cycle-consistency), which also serve as indicators of identity preservation (Monteiro et al., 2023). We choose conditional DDIMs for abduction and prediction, as this forms a bijection under the null intervention **pa** given a perfect noise estimator (Song et al., 2020b; Chao et al., 2023), resulting in good composition and reversibility. Existing works have used unconditional DDIM inversion or the aggregate inference posterior $q(\mathbf{x}_t|\mathbf{x}_0)$ for abduction followed by guidance for prediction (Komanduri et al., 2024; Sanchez et al., 2022a; Weng et al., 2023; Fang et al., 2024) (c.f. Section 3.3). In contrast, our spatial abduction enforces that spatial noise **u** will encode information pertaining to **pa** through conditioning, in line with Pearlian abduction.

Our spatial mechanism is trained using the standard denoising objective in Equation (5), setting $\mathbf{c} = \mathbf{pa}$. Training is more stable than VAE/HVAE mechanisms, which attempt to learn complex latent posteriors from data. Furthermore, through spatial abduction enabled by DDIM, we can significantly mitigate posterior-prior mismatch caused by projecting inputs into unsupported regions of low-dimensional latent space present in VAEs (Rezende & Viola, 2018; Ho et al., 2020; Ribeiro & Glocker, 2024). Instead, intermediate images $\mathbf{x}_{1:T-1}$ remain on the data manifold through iterative noise refinement by the denoiser. As such, **u** provides an in-distribution input to the prediction step with $h_\theta(\cdot)$.

### 3.2. Semantic Mechanism

Recall that the spatial noise **u** is iteratively refined by the denoiser during the counterfactual prediction process. As such, **u** alone cannot explicitly preserve the high-level semantics of **x** at each timestep (Preechakul et al., 2022). To address this, we introduce into our spatial mechanism a de-

codable high-level exogenous noise term **z**, which remains fixed during generation:

$$\mathbf{x} := f_\theta(\boldsymbol{\epsilon}, \mathbf{pa}) \approx h_\theta(\mathbf{u}, \mathbf{c}_{\text{sem}}) \quad (12)$$

where $\mathbf{c}_{\text{sem}} = (\mathbf{z}, \mathbf{pa})$, $h_\theta(\cdot)$ samples $p_\theta(\mathbf{x}|\mathbf{z}, \mathbf{pa})$, and exogenous noise $\boldsymbol{\epsilon}$ is decomposed independently into spatial **u** and semantic **z** terms $p(\boldsymbol{\epsilon}) = p(\mathbf{u})p(\mathbf{z})$ with $p(\mathbf{z}) = \mathcal{N}(\mathbf{0}, \boldsymbol{I})$. To learn $f_\theta(\cdot)$ such that **z** encodes high-level semantics, we marginalise $\int p_\theta(\mathbf{x}|\mathbf{z}, \mathbf{pa})d\mathbf{z}$ by introducing a semantic variational posterior $q_\varphi(\mathbf{z}|e_\varphi(\mathbf{x})) = \mathcal{N}(\mathbf{z}; \boldsymbol{\mu}_\varphi(\mathbf{x}), \boldsymbol{\sigma}_\varphi^2(\mathbf{x})\boldsymbol{I})$, where the probabilistic encoder $e_\varphi(\cdot)$ is implemented with a convolutional neural network (CNN). In practice, we use the surrogate objective:

$$\underset{\theta,\varphi}{\text{argmin}}\big\{\beta D_{\text{KL}}(q_\varphi(\mathbf{z}|e_\varphi(\mathbf{x}))|p(\mathbf{z})) + \quad (13)$$

$$\mathbb{E}_{\mathbf{x},\mathbf{c},t,\boldsymbol{\epsilon},\mathbf{z}}\|\boldsymbol{\epsilon} - \boldsymbol{\epsilon}_\theta(\hat{\mathbf{x}}_t, \mathbf{c}_{\text{sem}}, t)\|_2^2\big\}.$$

Here $\theta$ and $\varphi$ parametrise a conditional diffusion-based decoder and CNN-based encoder, respectively.

For semantic abduction, an approximation to the exogenous posterior can be defined using the probabilistic encoder in a manner similar to the amortised, explicit mechanism defined by Pawlowski et al. (2020):

$$p_{\mathfrak{S}}(\boldsymbol{\epsilon}|\mathbf{x}, \mathbf{pa}) = p_{\mathfrak{S}}(\mathbf{z}|\mathbf{x})p_{\mathfrak{S}}(\mathbf{u}|\mathbf{x}, \mathbf{c}_{\text{sem}})$$

$$\approx q_\varphi(\mathbf{z}|e_\varphi(\mathbf{x}))\delta(\mathbf{u} - h_\theta^{-1}(\mathbf{x}, \mathbf{c}_{\text{sem}})), \quad (14)$$

noting the key benefit of our model in that low-level noise is inferred via DDIM inversion $h_\theta^{-1}(\cdot)$ as opposed to a simple linear function (Section 3.1). Preechakul et al. (2022) presents a similar model (DiffAE) which trains a semantic encoder in unconstrained space, as such an additional diffusion model is trained post-hoc to sample **z** unconditionally for random image sampling. In contrast, our mechanism is

trained end-to-end with a regulariser, learning the semantic posterior $q_\varphi(\cdot)$ via variational inference, which we use for efficient semantic abduction. Image counterfactuals are now generated using Monte Carlo with $M$ particles,

$$\mathbf{z}^{(m)} \sim q_\varphi(\mathbf{z}|\boldsymbol{e}_\varphi(\mathbf{x})) \quad \mathbf{u}^{(m)} \approx h_\theta^{-1}(\mathbf{x}, \mathbf{c}_{\text{sem}}^{(m)})$$

$$\widetilde{\mathbf{x}} \approx \frac{1}{M} \sum_{m=1}^M h_\theta(\mathbf{u}^{(m)}, \widetilde{\mathbf{c}}_{\text{sem}}^{(m)}), \tag{15}$$

with $\mathbf{c}_{\text{sem}}^{(r)} = (\mathbf{z}^{(r)}, \mathbf{pa})$, $\widetilde{\mathbf{c}}_{\text{sem}}^{(r)} = (\mathbf{z}^{(r)}, \widetilde{\mathbf{pa}})$ and the semantic posterior sampled via the reparameterisation trick $\mathbf{z} = \boldsymbol{\mu}_\varphi(\mathbf{x}) + \boldsymbol{\sigma}_\varphi(\mathbf{x}) \odot \boldsymbol{\epsilon}$ with $\boldsymbol{\epsilon} \sim \mathcal{N}(\mathbf{0}, \boldsymbol{I})$. This depicted in Figure 1b. In Section 4, we generate deterministic image counterfactuals by using the degenerate semantic posterior during abduction such that $\mathbf{z} = \boldsymbol{\mu}_\varphi(\mathbf{x})$.

**Identifiability.** The unconditional VAE prior $p(\mathbf{z})$ in our model makes it non-identifiable (Locatello et al., 2019), meaning the true settings of $\theta$ and $\varphi$ cannot be uniquely determined even with infinite data. This can affect abduction, as multiple values of $\mathbf{z}$ may yield the same marginal likelihood $p_\theta(\mathbf{x}|\mathbf{pa})$, resulting in different counterfactuals under the same intervention. Khemakhem et al. (2020a) showed that identifiability can be improved by conditioning the semantic prior on observed variables, e.g., $p(\mathbf{z}|\mathbf{pa})$. Incorporating these ideas in our framework may require modifications akin to De Sousa Ribeiro et al. (2023), which we leave for future work.

### 3.3. Amortised, Anti-Causally Guided Mechanisms

The main limitation of conditional diffusion models is their tendency to ignore conditioning signals during generation. This is because noise injection during training often weakens the association between $\mathbf{c}$ and $\mathbf{x}$ (Dhariwal & Nichol, 2021; Nichol & Dhariwal, 2021), leading the model to rely on the shortcut $p_\theta(\mathbf{x}|\mathbf{c}) \approx p_\theta(\mathbf{x})$ to maximise likelihood (Chen et al., 2016). To address this, we incorporate *amortised guidance* into our mechanisms by reframing classifier-free guidance (Ho & Salimans, 2022) through a causal lens. This procedure is more parameter-efficient, easier to train, and encourages sample diversity (Dinh et al., 2023), compared to DiffSCM (Sanchez et al. (2022b)), who use a separately trained classifier for guidance (Dhariwal & Nichol, 2021).

We introduce amortised guidance into the semantic mechanism to enhance the counterfactual prediction step. We use a modified noise estimator within the denoiser (Equation (4)) derived from the sharpened, anti-causal score function $\nabla_\mathbf{x} \log p_{\mathfrak{S}}(\mathbf{c}_{\text{sem}}|\mathbf{x})^\omega$ (Appendix B.1):

$$\boldsymbol{\epsilon}_\theta(\mathbf{x}_t, \varnothing, t) + \omega(\boldsymbol{\epsilon}_\theta(\mathbf{x}_t, \widetilde{\mathbf{c}}_{\text{sem}}, t) - \boldsymbol{\epsilon}_\theta(\mathbf{x}_t, \varnothing, t)), \tag{16}$$

where $\boldsymbol{\epsilon}_\theta(\mathbf{x}_t, \varnothing, t)$ represents the unconditional noise estimate, $\varnothing$ is a guidance token introduced during training,

and the guidance scale $\omega > 1$ amplifies the counterfactual-conditioned noise estimate. For $\boldsymbol{\epsilon}_\theta(\cdot)$ to amortise conditional and unconditional representations, $\varnothing$ replaces $\mathbf{c}_{\text{sem}}$ in the denoising objective (Equation (13)) with a probability $p_\varnothing$. Our choice of $p_\varnothing$ is tuned for counterfactual soundness ($\mathcal{L}_3$), as opposed to most existing works, who choose $p_\varnothing$ for diverse $\mathcal{L}_2$-level sampling (Ho & Salimans, 2022).

The amortised, anti-causally guided semantic mechanism is

$$\mathbf{x} := f_\theta(\boldsymbol{\epsilon}, \mathbf{pa}) \approx h_\theta^\omega(\mathbf{u}, \mathbf{c}_{\text{sem}} \cup \{\varnothing\})$$
$$:= (h_{1|\mathfrak{C}}^\omega \circ \cdots \circ h_{T|\mathfrak{C}}^\omega)(\mathbf{u}), \tag{17}$$

where $h_\theta^\omega(\cdot)$ denotes DDIM using the modified noise estimator (Equation (16)) and $\mathfrak{C} = \{\mathbf{c}_{\text{sem}}, \varnothing, \theta\}$. Counterfactual generation follows:

$$\mathbf{z} \sim q_\varphi(\mathbf{z}|\boldsymbol{e}_\varphi(\mathbf{x})), \quad \mathbf{u} \approx h_\theta^{-1}(\mathbf{x}, \mathbf{c}_{\text{sem}}),$$
$$\widetilde{\mathbf{x}} \approx h_\theta^\omega(\mathbf{u}, \widetilde{\mathbf{c}}_{\text{sem}} \cup \{\varnothing\}). \tag{18}$$

Amortised, anti-causal guidance can also be used with our spatial mechanism by using $\nabla_\mathbf{x} \log p_{\mathfrak{S}}(\mathbf{pa}|\mathbf{x})^\omega$ to derive the noise estimator (Equation (16)). In this case, counterfactual generation becomes

$$\mathbf{u} \approx h_\theta^{-1}(\mathbf{x}, \mathbf{pa}), \quad \widetilde{\mathbf{x}} \approx h_\theta^\omega(\mathbf{u}, \widetilde{\mathbf{pa}} \cup \{\varnothing\}), \tag{19}$$

The option to use guidance with both mechanisms is depicted in Figures 1a and 1b, via blue edges.

**Dynamic Semantic Abduction.** Recall that counterfactual soundness is assessed through *composition* (reconstruction) and *effectiveness* (interventional faithfulness). Amortised anti-causal guidance ensures sound effectiveness by boosting counterfactual-conditioned noise estimates. However, high guidance scales may suppress unconditional estimates, compromising composition and, therefore, identity preservation. Analogous tradeoffs are observed in diffusion-based image editing (Yang et al., 2024; Song et al., 2024; Tang et al., 2024) with text-conditional latent diffusion. Notably, Mokady et al. (2023) notice that DDIM inversion produces poor reconstructions when amortised guidance is used for editing with Stable Diffusion (Rombach et al., 2022). They address this by optimising guidance tokens at each timestep to align the inverse and guided trajectories.

Building on this idea, we propose *counterfactual trajectory alignment* (CTA) to improve composition by tuning guidance tokens, now modelled as exogenous noise terms. We find that a linear update for guidance tokens for each image at each timestep (from $T$ to 1) is more computationally efficient than Mokady et al. (2023); Yang et al. (2024) and sufficient to improve identity preservation:

$$\varnothing_t^\star \leftarrow \text{CTA}\left(\varnothing_t^\star, \mathbf{x}_{t-1}, \mathbf{x}_{t-1}^\omega\right)$$
$$:= \varnothing_t^\star - \eta \nabla_{\varnothing_t^\star} \|\mathbf{x}_{t-1} - \mathbf{x}_{t-1}^\omega\|_2^2, \tag{20}$$

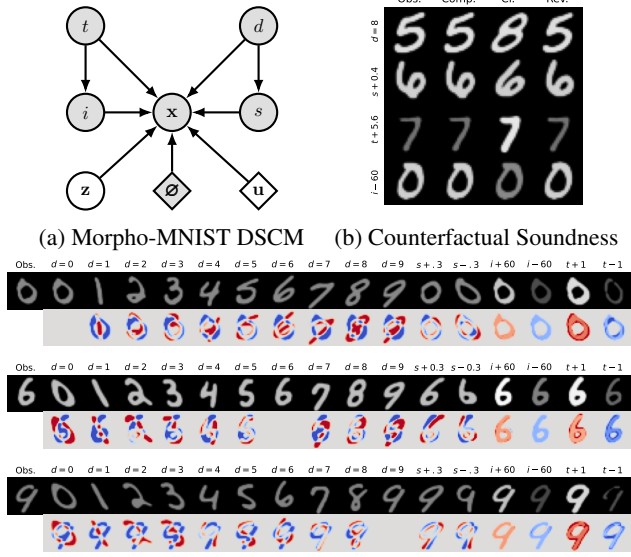

(a) Morpho-MNIST DSCM  (b) Counterfactual Soundness

(c) Morpho-MNIST Image Counterfactuals

*Figure 2.* Morpho-MNIST ($28 \times 28$) counterfactuals generated using an amortised, anti-causally guided semantic mechanism ($p_\varnothing = 0.1, \omega = 1.5$) based on the DSCM shown in (a). (b) illustrates counterfactual soundness (Obs: Observation, Comp: Composition, Cf: Counterfactual, Rev: Reversibility). (c) depicts image counterfactuals: interventions are shown above the top row and the bottom row visualises total causal effects (red: increase, blue: decrease), refer to (Appendix A.2) for details.

where $\varnothing_T^\star$ initialised to $\varnothing$, $\varnothing_{t-1}^\star \leftarrow \varnothing_t^\star$ is set at the end of each timestep, $\mathbf{x}_{t-1}^\omega = h_{t|\mathfrak{C}'}^\omega(\mathbf{x}_t^\omega)$ with $\mathfrak{C}' = \{\mathbf{c}_{\text{sem}}, \varnothing_t^\star, \theta\}$, $\mathbf{x}_{t-1} = h_{t-1|\mathfrak{C}}^{-1}(\mathbf{x}_{t-2})$ with $\mathfrak{C} = \{\mathbf{c}_{\text{sem}}, \theta\}$, $\eta$ is the step size and we use $\omega > 1$. We provide the full algorithm in Appendix B.2. We use CTA within *dynamic* semantic abduction for our guided semantic mechanism. Guidance tokens are modelled as independent exogenous noise terms via $p(\boldsymbol{\epsilon}) = p(\mathbf{z})p(\mathbf{u})\prod_{t=1:T} \delta(\varnothing_t - \varnothing)$. The exogenous posterior is approximated by $p_\mathfrak{S}(\boldsymbol{\epsilon}|\mathbf{x}, \mathbf{pa})$

$$= p_\mathfrak{S}(\mathbf{z}|\mathbf{x})p_\mathfrak{S}(\mathbf{u}|\mathbf{x}, \mathbf{c}_{\text{sem}}) \prod_{t=1}^T p_\mathfrak{S}(\varnothing_t|\mathbf{x}_t, \mathbf{x}_t^\omega) \qquad (21)$$

$$\approx q_\varphi(\mathbf{z}|\mathbf{e}_\varphi(\mathbf{x}))\delta(\mathbf{u} - h_\theta^{-1}(\mathbf{x}, \mathbf{c}_{\text{sem}})) \prod_{t=1}^T \delta(\varnothing_t - \varnothing_t^\star),$$

and counterfactuals are generated as

$$\widetilde{\mathbf{x}} \approx h_\theta^\omega(\mathbf{u}, \widetilde{\mathbf{c}}_{\text{sem}} \cup \{\varnothing_{1:T}^\star\})$$
$$:= (h_{1|\mathfrak{C}_1}^\omega \circ \cdots \circ h_{T|\mathfrak{C}_T}^\omega)(\mathbf{u}), \qquad (22)$$

where $\mathfrak{C}_t = \{\widetilde{\mathbf{c}}_{\text{sem}}, \varnothing_t^\star, \theta\}$. This is similar to Equation (18) with $\varnothing$ set to the result of CTA at each timestep.

*Table 1.* Soundness of Morpho-MNIST image counterfactuals generated under $do(d)$ from DSCMs modelling the relationship $d \to \mathbf{x}$, in which the digit class ($d$) is the only parent of the image ($\mathbf{x}$), with data generated from the true SCM in Appendix E. Amortised, anti-causally guided mechanisms are trained with $p_\varnothing = 0.1$. Effectiveness (EFF.) is measured by the accuracy (Acc) of a pretrained classifier. Diffusion counterfactuals are normalised to $[0, 1]$ to measure composition (COMP.) and reversibility (REV.) while ensuring faithful comparison to VAE/HVAE baselines.

| MECHANISM | COMP. $L_1 \downarrow (\times 10^{-2})$ | EFF. Acc $\uparrow$ | REV. $L_1 \downarrow (\times 10^{-2})$ |
|---|---|---|---|
| DIFFSCM (Sanchez & Tsaftaris, 2022) | 0.410 | 17.02 | 1.31 |
| VCI (Wu et al., 2024) | 2.05 | 92.48 | 6.71 |
| SPATIAL $\{\omega = 1.5\}$ | 0.615 | 99.63 | 2.56 |
| SPATIAL $\{\omega = 3\}$ | 1.92 | 99.95 | 3.42 |
| SEMANTIC $\{\omega = 1.5\}$ | 0.342 | 97.46 | 2.53 |
| SEMANTIC $\{\omega = 3\}$ | 1.20 | 99.90 | 3.17 |

## 4. Experiments

We present three case studies using our mechanisms for counterfactual image generation [1]. We begin with a toy scenario where we control the true causal data-generating process, and progressively scale up our mechanisms for causal face modelling and a novel medical artefact removal problem. We compare our mechanisms against VAE, HVAE and diffusion-based alternatives (Pawlowski et al., 2020; De Sousa Ribeiro et al., 2023; Wu et al., 2024; Sanchez & Tsaftaris, 2022) using counterfactual soundness metrics.

**Morpho-MNIST** We begin by applying our proposed mechanisms within a DSCM to a Morpho-MNIST dataset (Castro et al., 2019) generated from a known SCM (Appendix E). The corresponding computational graph, shown in Figure 2a, extends the work of (De Sousa Ribeiro et al., 2023) to introduce a challenging causal relationship between digit class ($d$) and slant ($s$). We also use this dataset to implement DSCMs modelling a subset of mechanisms from the true SCM, $\{d \to \mathbf{x}\}$ and $\{t \to i, t \to \mathbf{x} \leftarrow i\}$, with results for the latter presented in Appendix E.2. Additionally, we construct an SCM for a colourised variant of Morpho-MNIST, where the digit class ($d$) causes hue ($h$), presented in Appendix F. We use the true-data generating mechanisms for $\mathbf{pa}$ within these DSCMs. Effectiveness is measured using a pre-trained classifier for $d$ and measurement functions provided by the Morpho-MNIST library for $i$, $s$ and $t$.

Table 1 reports the counterfactual soundness results for simple DSCMs modelling only the mechanism $d \to \mathbf{x}$, assessed under random interventions $do(d)$. Our amortised, anti-causally guided mechanisms outperform VCI in both effectiveness and identity preservation, as measured by composition and reversibility. Notably, semantic mechanisms achieve better identity preservation than their spatial counterparts for the same $\omega$, albeit with minor reductions in ef-

---

[1] https://github.com/RajatRasal/Diffusion-Counterfactuals

*Table 2.* Soundness of Morpho-MNIST image counterfactuals under $do(s)$ and $do(d)$ using DSCMs modelling the SCM in Appendix E. Effectiveness for digit class ($d$) is measured using accuracy (Acc) from a pre-trained classifier and mean absolute percentage error (MAPE) for slant ($s$), thickness ($t$) and intensity ($i$). Counterfactuals are normalised to $[0, 1]$ to measure composition (COMP.) and reversibility (REV.). Metrics are scaled by $\times 10^{-2}$, except for MAPE (s), which is scaled by $\times 10^{-1}$, and Acc ($d$), which remains unscaled.

| | SLANT INTERVENTION ($do(s)$) | | | | | CLASS INTERVENTION ($do(d)$) | | | | | NULL |
| | EFFECTIVENESS | | | | REV. | EFFECTIVENESS | | | | REV. | COMP. |
| MECHANISM | MAPE ($t$) ↓ | MAPE ($i$) ↓ | MAPE ($s$) ↓ | Acc ($d$) ↑ | $L_1$ ↓ | MAPE ($t$) ↓ | MAPE ($i$) ↓ | MAPE ($s$) ↓ | Acc ($d$) ↑ | $L_1$ ↓ | $L_1$ ↓ |
|---|---|---|---|---|---|---|---|---|---|---|---|
| VAE (Pawlowski et al., 2020) | 4.63 | 6.98 | 2.91 | 97.27 | 2.14 | 5.90 | 8.03 | 2.10 | 94.92 | 2.44 | 1.81 |
| HVAE (De Sousa Ribeiro et al., 2023) | 3.39 | 0.493 | 3.88 | 95.02 | 0.615 | 4.39 | 0.508 | 1.23 | 95.31 | 1.62 | 0.008 |
| VCI (Wu et al., 2024) | 3.08 | 0.652 | 0.907 | 90.04 | 1.60 | 2.63 | 0.632 | 0.912 | 94.62 | 0.990 | 0.655 |
| SPATIAL: | 2.78 | 0.552 | 2.45 | 96.62 | 2.74 | 3.11 | 0.592 | 2.13 | 96.29 | 3.78 | 0.555 |
| $\{\omega = 1.5, \, p_\varnothing = 0.1\}$ | 2.17 | 0.591 | 1.53 | 99.02 | 3.52 | 2.26 | 0.546 | 0.501 | 99.51 | 4.50 | 2.02 |
| $\{\omega = 3, \, p_\varnothing = 0.1\}$ | 1.85 | 0.291 | 0.885 | 99.84 | 5.25 | 1.87 | 0.292 | 0.855 | 99.90 | 5.70 | 3.29 |
| $\{\omega = 4.5, \, p_\varnothing = 0.1\}$ | 1.87 | 0.306 | 0.667 | 99.90 | 5.95 | 1.85 | 3.19 | 0.511 | 99.98 | 6.30 | 3.71 |
| $\{\omega = 1.5, \, p_\varnothing = 0.5\}$ | 2.88 | 0.739 | 2.43 | 97.75 | 2.94 | 3.44 | 0.802 | 1.29 | 93.95 | 4.31 | 0.660 |
| $\{\omega = 3, \, p_\varnothing = 0.5\}$ | 2.01 | 0.388 | 0.984 | 99.64 | 4.18 | 2.55 | 0.445 | 0.932 | 97.63 | 4.93 | 1.71 |
| $\{\omega = 4.5, \, p_\varnothing = 0.5\}$ | 2.02 | 0.393 | 1.84 | 99.71 | 4.49 | 2.48 | 0.460 | 0.783 | 98.14 | 5.30 | 1.77 |
| SEMANTIC: | 4.98 | 0.981 | 4.79 | 90.33 | 3.94 | 7.43 | 1.63 | 7.15 | 83.01 | 5.60 | 0.139 |
| $\{\omega = 1.5, \, p_\varnothing = 0.1\}$ | 2.83 | 1.20 | 1.51 | 98.44 | 2.98 | 3.88 | 1.13 | 1.43 | 97.66 | 4.33 | 0.940 |
| $\{\omega = 3, \, p_\varnothing = 0.1\}$ | 2.14 | 1.00 | 1.48 | 99.80 | 4.41 | 2.15 | 0.941 | 0.699 | 99.80 | 5.35 | 2.17 |
| $\{\omega = 4.5, \, p_\varnothing = 0.1\}$ | 2.13 | 0.796 | 1.78 | 99.80 | 5.35 | 7.67 | 0.941 | 0.762 | 99.93 | 6.15 | 2.87 |

fectiveness. DiffSCM struggles to generate counterfactuals that faithfully reflect the intervention, as such they attain the best identity preservation because of poor effectiveness. We present strategies to improve DiffSCM's effectiveness and their trade-offs with identity preservation in Appendix E.3.

Figure 2c illustrates that our mechanisms can model complex causal relationships: increasing $t$ and using larger values of $d$ result in increases in $i$ and $s$, respectively, while $i$ and $s$ can be controlled independently of their parents. Counterfactual soundness results, seen in Figure 2b, are summarised in Table 2 for random interventions $do(s)$ and $do(d)$, with additional results for $do(t)$ and $do(i)$ in Appendix E.2. Our mechanisms perform comparably or better than baselines, with guidance improving effectiveness beyond them. Increasing $\omega$ improves effectiveness at the cost of identity preservation (composition and reversibility). For a chosen $\omega$, mechanisms trained with $p_\varnothing = 0.5$ see a small drop in effectiveness compared to those trained with $p_\varnothing = 0.1$, while exhibiting better identity preservation. This improvement is attributed to reduced sample diversity at higher $p_\varnothing$ (Ho & Salimans, 2022), which we deem advantageous for image editing tasks. Notably, for the same $\omega$ and $p_\varnothing$, guided semantic mechanisms achieve comparable effectiveness to guided spatial mechanisms across many confounders of **x** while substantially improving identity preservation. While VCI achieves good MAPE on $t$, $i$, and $s$ under $do(s)$ interventions, it exhibits significantly lower $d$ accuracy here compared to many ablations of our proposed mechanisms. This suggests that, while localised interventions are faithfully obeyed, global edits to digit class remain challenging. In contrast, our guided mechanisms achieve higher $d$ accuracy, with comparable effectiveness for other covariates, demonstrating a more balanced trade-off between intervention faithfulness and preservation of core image characteristics.

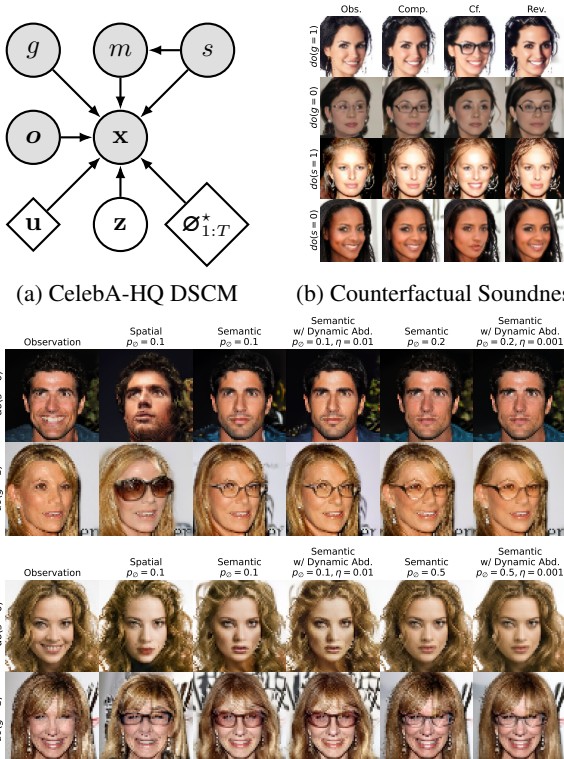

(a) CelebA-HQ DSCM          (b) Counterfactual Soundness

(c) Semantic Abduction improves Identity Preservation

*Figure 3.* CelebA-HQ ($64 \times 64$) counterfactuals generated using amortised, anti-causally guided semantic mechanisms. (a) DSCM with spatial (**u**), semantic (**z**) and dynamic ($\varnothing_{1:T}^\star$) exogenous noise terms for **x**. (b) shows counterfactual soundness using semantic abduction with $p_\varnothing = 0.1$ and $\omega = 2$. (c) shows that semantic mechanisms improve identity preservation, and dynamic abduction further improves backgrounds, hairstyle, skin colour and facial structure. Here, the choice of $\eta$ is fine-tuned for each observation.

*Table 3.* Soundness of CelebA-HQ image counterfactuals generated using our proposed diffusion-based mechanisms under simulated interventions. Effectiveness is measured using the F1-scores from pre-trained classifiers for eyeglasses ($g$) and smiling ($s$).

| | Eyeglasses Intervention ($do(g)$) | | | | Smiling Intervention ($do(s)$) | | | | Null |
|---|---|---|---|---|---|---|---|---|---|
| | Effectiveness | | Rev. | IDP | Effectiveness | | Rev. | IDP | Comp. |
| Mechanism | F1($s$) ↑ | F1($g$) ↑ | $L_1$ ↓ | LPIPS ↓ | F1($s$) ↑ | F1($g$) ↑ | $L_1$ ↓ | LPIPS ↓ | $L_1$ ↓ |
| Spatial: | 95.65 (0.12) | 94.08 (0.24) | 0.084 (0.0004) | 0.119 (0.0002) | 95.65 (0.04) | 95.32 (0.78) | 0.078 (0.0003) | 0.102 (0.0003) | 0.034 (0.0005) |
| {$\omega$=2, $p_\varnothing$=0.1} | 98.33 (0.00) | 99.07 (0.03) | 0.211 (0.0003) | 0.171 (0.0002) | 99.09 (0.12) | 99.23 (0.00) | 0.183 (0.0003) | 0.139 (0.0007) | 0.130 (0.0004) |
| {$\omega$=3, $p_\varnothing$=0.1} | 98.34 (0.25) | 99.27 (0.11) | 0.297 (0.0006) | 0.197 (0.0009) | 99.59 (0.16) | 99.19 (0.00) | 0.264 (0.0004) | 0.161 (0.0005) | 0.196 (0.0004) |
| {$\omega$=2, $p_\varnothing$=0.5} | 97.12 (0.35) | 93.85 (0.13) | 0.141 (0.0004) | 0.138 (0.0009) | 96.51 (0.01) | 96.30 (0.38) | 0.127 (0.0002) | 0.110 (0.0003) | 0.090 (0.0001) |
| {$\omega$=3, $p_\varnothing$=0.5} | 97.73 (0.30) | 96.33 (0.15) | 0.214 (0.0001) | 0.161 (0.0007) | 98.33 (0.16) | 98.12 (0.37) | 0.188 (0.0002) | 0.128 (0.0004) | 0.139 (0.0001) |
| Semantic: | 93.54 (0.14) | 93.31 (0.18) | 0.048 (0.0001) | 0.077 (0.0000) | 92.25 (0.06) | 95.80 (0.00) | 0.039 (0.0001) | 0.050 (0.0000) | 0.007 (0.0002) |
| {$\omega$=2, $p_\varnothing$=0.1} | 99.09 (0.10) | 96.86 (0.03) | 0.185 (0.0001) | 0.096 (0.0002) | 94.93 (0.15) | 99.45 (0.35) | 0.183 (0.0002) | 0.066 (0.0002) | 0.130 (0.0001) |
| {$\omega$=3, $p_\varnothing$=0.1} | 99.33 (0.13) | 97.67 (0.05) | 0.222 (0.0002) | 0.107 (0.0002) | 95.74 (0.24) | 99.47 (0.38) | 0.220 (0.0002) | 0.077 (0.0001) | 0.174 (0.0001) |
| {$\omega$=2, $p_\varnothing$=0.2} | 98.46 (0.20) | 95.82 (0.12) | 0.181 (0.0001) | 0.102 (0.0002) | 94.08 (0.35) | 99.73 (0.38) | 0.177 (0.0002) | 0.066 (0.0002) | 0.122 (0.0002) |
| {$\omega$=3, $p_\varnothing$=0.2} | 98.73 (0.17) | 97.53 (0.18) | 0.224 (0.0001) | 0.114 (0.0004) | 95.19 (0.12) | 99.75 (0.34) | 0.221 (0.0002) | 0.080 (0.0002) | 0.168 (0.0003) |

*Figure 4.* CelebA-HQ: Effect of guidance scale ($\omega$), depicted by the number on each dot, on composition, model identity preservation (IDP) and effectiveness of amortised anti-causally guided mechanisms trained with $p_\varnothing = 0.1$.

*Figure 5.* CelebA-HQ: Effect of step size ($\eta$) and guidance scale ($\omega$) on model identity preservation (IDP) when using dynamic abduction with amortised anti-causally guided semantic mechanism ($p_\varnothing = 0.1$) on 200 randomly chosen images from the val. set.

**CelebA-HQ** We now scale-up our mechanisms for causal modelling of real-world images in CelebA-HQ (Karras, 2017). Here, we model the binary variables *Smiling* ($s$), *Mouth Open* ($m$), and *Eyeglasses* ($g$) as parents of $\mathbf{x}$, and $s \rightarrow m$, as shown by the computational graph in Figure 3a. We conduct interventions on $g$ by toggling its observed value, $do(g')$, and simulate interventions on $s$ with $do(s') = do(s = s', m = s')$, reflecting associative $\mathcal{L}_1$-level statistics in CelebA-HQ and our knowledge of facial expressions, i.e. smiling causes the mouth to open. In preliminary experiments, we observed that spurious correlations under $do(s')$ sometimes led to unintended effects like hair loss or removal of glasses. To mitigate this, we concatenate confounding attributes – *Male*, *Wearing Lipstick*, *Bald*, and *Wearing Hat* – into a variable $\mathbf{o}$, modelled as an independent parent of $\mathbf{x}$, which we do not subject to interventions. We discuss baseline models in Appendix H.2.

The counterfactual soundness trends observed in Morpho-MNIST, due to the choice of $p_\varnothing$ and $\omega$, are further amplified in CelebA-HQ, with results summarised in Table 3 and visualisations in Figure 3b. In spatial mechanisms, setting $p_\varnothing = 0.5$ improves reversibility more under $do(g)$ than the localised intervention $do(s)$, but at the cost of lower effectiveness, as reflected in the reduced F1-scores. In semantic mechanisms, increasing $p_\varnothing$ from 0.1 to 0.2 yields a slight improvement in composition, while reversibility and IDP remain comparable. Notably, semantic mechanisms improve identity preservation metrics over their spatial counterparts with the same $p_\varnothing$ and $\omega$, by better preserving high-level facial characteristics as shown in Figure 3c, albeit incurring a small drop in effectiveness. Figure 4 further illustrates the effect of $\omega$ on identity preservation and effectiveness. Specifically, IDP and composition increase linearly with $\omega$, indicating a decrease in identity preservation, while effectiveness increases across both mechanisms. Importantly, the semantic mechanism consistently outperforms the spatial mechanism in identity preservation across all values of $\omega$.

Dynamic semantic abduction improves the preservation of intricate facial attributes, such as hairstyle, skin tone/illumination, facial structure, and backgrounds, as illustrated in Figure 3c. Figure 5 demonstrates the effect of step size ($\eta$) in CTA, revealing that global interventions such as $do(g)$ may require a larger $\eta$ to improve IDP compared to localised interventions $do(s)$. Moreover, effectiveness improves alongside IDP in CTA. However, these improvements come with additional computational cost: dynamic semantic abduction requires $\sim$ 3 minutes per image, compared to $\sim$ 3 minutes and $\sim$ 3.5 minutes for the guided spatial and semantic mechanisms, respectively, using a batch size of 128 on an NVIDIA GeForce RTX 4090.

**EMBED** Using prior insights, we apply our mechanisms to a real-world artefact removal task on the EMory BrEast imaging Dataset (EMBED) (Jeong et al., 2022). Schueppert et al. (2024) observe that triangular and circular skin markers are spuriously associated with breast cancer in classifiers due to shortcut learning (Geirhos et al., 2020), and manually labelled 22,012 affected mammograms. Using this dataset, we train a significantly scaled-up, amortised, anti-causally guided semantic mechanism ($p_\varnothing = 0.1, \omega = 1.2$) to remove skin markers. We model triangular markers ($t$), circular markers ($c$), breast density ($d$), and cancer ($y$) as independent parents of the mammogram $\mathbf{x}$, and remove artefacts by intervening on $t$ and $c$ while holding $d$ and $y$ fixed. Figure 6 shows that our mechanisms effectively remove artefacts and can disentangle representations for triangles and circles. We successfully remove **$95.16 \pm 1.34\%$** of triangles and **$91.69 \pm 1.06\%$** of circles in our test set - a noteworthy result given the dataset's small size and the scarcity of labelled skin markers (Appendix I).

## Conclusion

Our work highlights trade-offs inherent in using diffusion models, studied extensively on associative ($\mathcal{L}_1$) and interventional ($\mathcal{L}_2$) applications, for counterfactual inference ($\mathcal{L}_3$). We introduce diffusion-based causal mechanisms with semantic abduction capabilities and demonstrate their enhanced identity preservation in image counterfactuals, as measured by counterfactual soundness metrics. Notably, we show that large $\omega$ and smaller $p_\varnothing$, popular at $\mathcal{L}_2$, compromise identity preservation (composition and reversibility) in favour of causal control (effectiveness) at $\mathcal{L}_3$. Current diffusion approaches are biased towards $\mathcal{L}_2$, limiting their soundness at $\mathcal{L}_3$ and widespread adoption at $\mathcal{L}_1$. Since $\mathcal{L}_3$ subsumes $\mathcal{L}_1$ and $\mathcal{L}_2$, we argue that a causal lens on diffusion modelling at $\mathcal{L}_3$ is essential for tackling a broad range of problems, rather than relying on models optimised for random sampling at $\mathcal{L}_2$. This would better generalise diffusion models to real-world tasks that demand creative reasoning and causal understanding - where diffusion already excels.

Our findings present new opportunities for using diffusion models at $\mathcal{L}_3$. For example, fast generation with spatial mechanisms could enable efficient counterfactual data augmentation (Roschewitz et al., 2024). Enhanced identity preservation via semantic mechanisms could improve counterfactual explanations for downstream models (Augustin et al., 2022). Dynamic abduction could support stress testing by providing higher image fidelity for challenging test cases (Pérez-García et al., 2023). Lastly, these mechanisms could inspire causally-informed generative recognition models, integrating causal representations into the human-aligned perceptual capabilities of diffusion models (Jaini et al., 2023).

**Limitations** This work uses high-quality but small datasets,

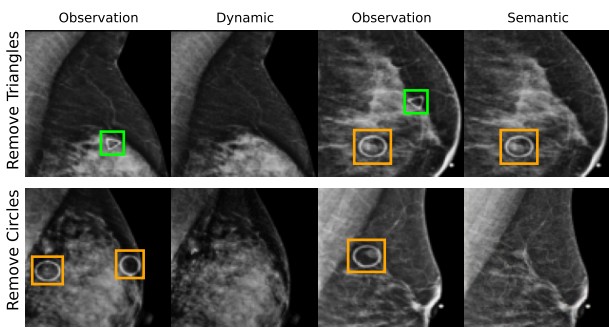

*Figure 6.* Skin marker removal on EMBED ($192 \times 192$) using an amortised, anti-causally guided semantic mechanism with and without dynamic abduction; (orange: circle, green: triangle).

which limits our mechanisms' ability to learn robust strided trajectories, resulting in slower counterfactual inference. Future work could address this by using larger datasets to enable better strided generation and distillation techniques (Salimans & Ho, 2022; Song et al., 2023) to improve efficiency. Our focus on Markovian SCMs assumes that causal effects are identifiable, however, this assumption need not hold in practice. Noisy labels (Lingenfelter et al., 2022; Thyagarajan et al., 2022) and low-resolution images can challenge the assumptions of our causal graph, potentially affecting the learned representations under untested interventions. Future work could tackle assessing the impact of noisy labels in a controlled setting or using corrected labels where available (Wu et al., 2023). Additionally, we do not guarantee that $\mathbf{pa} \perp\!\!\!\perp \mathbf{z}|\mathbf{x}$, but instead set the dimensionality of $\mathbf{z}$ and value of $\beta$ such that we can control $\mathbf{pa}$ whilst improving identity preservation. Incorporating techniques for model identifiability (Khemakhem et al., 2020a; Yan et al., 2023; De Sousa Ribeiro et al., 2023) and providing guarantees on the contents of $\mathbf{z}$ (Chen et al., 2025; Von Kügelgen et al., 2021) present useful directions for future research.

## Acknowledgements

We thank Pavithra Manoj, Mélanie Roschewitz, and Michael Tänzer for their detailed and insightful feedback on early versions of this manuscript. R.R. is supported by the Engineering and Physical Sciences Research Council (EPSRC) through a Doctoral Training Partnerships PhD Scholarship. A.K. is supported by UKRI (grant no. EP/S023356/1), as part of the UKRI Centre for Doctoral Training in Safe and Trusted AI. B.G. received support from the Royal Academy of Engineering as part of his Kheiron/RAEng Research Chair. B.G. and F.R. acknowledge the support of the UKRI AI programme, and the EPSRC, for CHAI - EPSRC Causality in Healthcare AI Hub (grant no. EP/Y028856/1).

## Impact Statement

This paper demonstrates the trade-offs inherent in using diffusion models, widely optimised for $\mathcal{L}_2$-level random sampling, to perform $\mathcal{L}_3$-level counterfactual inference. By uncovering these trade-offs, our work demonstrates the need to evaluate diffusion models from a perspective that considers fairness and mitigates spurious correlations. Counterfactual inference shares strong similarities with text-driven image-to-image translation and editing, and our findings encourage practitioners to rethink the design of foundational diffusion models, which underpin these applications, moving beyond a focus on high-quality random sampling. Notably, in latent diffusion models, text control alone often fails to account for causal dependencies, instead defaulting to biases learned from training data, which can result in unintended consequences. In text-guided image editing, prompts alone cannot be used to infer causal structure, leaving the model to infer relationships based solely on correlations observed in training data. As a result, even minor edits can produce spurious changes. Adopting a structured causal perspective - where models implicitly discover or explicitly learn SCMs - enables developers to transparently state their model's assumptions and update them as new information becomes available through the causal graph, and restricts misuse by enforcing causal rules. These considerations are critical in real-world applications like face modelling and medical imaging, where dataset biases can lead to flawed decision-making. In such high-stakes scenarios, ensuring that models are interpretable and transparent is essential, as black-box models often fail to provide the clarity needed to justify their outputs.

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

# A. Background

## A.1. Evaluating Counterfactuals

To formalise counterfactual soundness metrics, we use the notion of an image counterfactual function $\mathcal{F}_\theta(\cdot)$, which generates counterfactuals as $\widetilde{\mathbf{x}} := \mathcal{F}_\theta(\mathbf{x}, \mathbf{pa}, \widetilde{\mathbf{pa}}) = f_\theta(f_\theta^{-1}(\mathbf{x}, \mathbf{pa}), \widetilde{\mathbf{pa}})$. Composition measures $L_1$-reconstruction error by comparing observed images with counterfactuals under null interventions,

$$\mathrm{Comp}(\mathbf{x}, \mathbf{pa}) := L_1(\mathbf{x}, \mathcal{F}_\theta(\mathbf{x}, \mathbf{pa}, \mathbf{pa})). \tag{23}$$

Reversibility assesses cycle-consistency via the $L_1$-distance between the observed image and its cycled-back counterfactual,

$$\mathrm{Rev}(\mathbf{x}, \mathbf{pa}, \widetilde{\mathbf{pa}}) := L_1(\mathbf{x}, \mathcal{F}_\theta(\mathcal{F}_\theta(\mathbf{x}, \mathbf{pa}, \widetilde{\mathbf{pa}}), \widetilde{\mathbf{pa}}, \mathbf{pa})). \tag{24}$$

Effectiveness quantifies faithfulness to interventions. We use a metric $L_k(\cdot)$ to compute the error between the counterfactual parent $\widetilde{\mathbf{pa}}_k$ and the output of an anti-causal parent predictor $p(\mathbf{pa}_k|\mathbf{x})$ sampled using the function $\mathbf{Pa}_k(\mathbf{x})$,

$$\mathrm{Eff}(\mathbf{x}, \mathbf{pa}, \widetilde{\mathbf{pa}}) := L_k(\widetilde{\mathbf{pa}}_k, \mathbf{Pa}_k(\mathcal{F}_\theta(\mathbf{x}, \mathbf{pa}, \widetilde{\mathbf{pa}}))). \tag{25}$$

## A.2. Causal Mediation Analysis

The general causal effect is defined as

$$\mathrm{CE}(\mathbf{x}, \widetilde{\mathbf{x}}, \mathbf{pa}) := \widetilde{\mathbf{x}} - \mathcal{F}_\theta(\mathbf{x}, \mathbf{pa}, \mathbf{pa}), \tag{26}$$

where $\widetilde{\mathbf{x}}$ is varied in order to compute the *direct*, *indirect* and *total* causal effects. Following (Pearl, 2001), we define the direct effect using

$$\widetilde{\mathbf{x}}_{\mathrm{DE}} = \mathcal{F}_\theta(\mathbf{x}, \mathbf{pa}, \widetilde{\mathbf{pa}}_{\mathrm{DE}}), \quad \widetilde{\mathbf{pa}}_{\mathrm{DE}} = (\mathbf{pa}\backslash\{\mathbf{pa}_k\}) \cup \{\widetilde{\mathbf{pa}}_k\}, \tag{27}$$

where mediators remain fixed to their observed values, and only $\mathbf{pa}_k$ is modified. The indirect effect is computed as

$$\widetilde{\mathbf{x}}_{\mathrm{IDE}} = \mathcal{F}_\theta(\mathbf{x}, \mathbf{pa}, \widetilde{\mathbf{pa}}_{\mathrm{IDE}}), \quad \widetilde{\mathbf{pa}}_{\mathrm{IDE}} = (\widetilde{\mathbf{pa}}\backslash\{\widetilde{\mathbf{pa}}_k\}) \cup \{\mathbf{pa}_k\} \tag{28}$$

where mediators are set according to $do(\mathbf{pa}_k)$ while $\mathbf{pa}_k$ retains its observed value. The total causal effect combines both direct and indirect effects,

$$\widetilde{\mathbf{x}}_{\mathrm{TE}} = \mathcal{F}_\theta(\mathbf{x}, \mathbf{pa}, \widetilde{\mathbf{pa}}). \tag{29}$$

We verify our computations using the fact that $\widetilde{\mathbf{x}}_{\mathrm{DE}} + \widetilde{\mathbf{x}}_{\mathrm{IDE}} = \widetilde{\mathbf{x}}_{\mathrm{TE}}$. For our purposes, this framework provides qualitative insights for debugging and explaining the model's causal behaviour.

# B. Methods

## B.1. Guided Counterfactual Prediction Step

$$
\begin{aligned}
\nabla_{\mathbf{x}_t} \log p(\mathbf{pa}|\mathbf{x}_t)^\omega &= \omega \nabla_{\mathbf{x}_t} \log p(\mathbf{pa}|\mathbf{x}_t) \\
&= \omega(\nabla_{\mathbf{x}_t} \log p(\mathbf{x}_t|\mathbf{pa}) - \nabla_{\mathbf{x}_t} \log p(\mathbf{x}_t)) \\
&\propto \omega(\boldsymbol{\epsilon}(\mathbf{x}_t, t, \mathbf{pa}) - \boldsymbol{\epsilon}(\mathbf{x}_t, t, \varnothing)),
\end{aligned}
\tag{30}
$$

## B.2. Counterfactual Trajectory Alignment

---

**Algorithm 1** Counterfactual Trajectory Alignment

---

1: **Input:** Images $[\mathbf{x}_T = \mathbf{u}, ..., \mathbf{x}_0 = \mathbf{x}]$ from $h_\theta^{-1}(\mathbf{x}, \mathbf{c}_{\text{sem}})$, step size $\eta$, guidance scale $\omega > 1$, guidance token $\varnothing$.
2: **Output:** Optimised exogenous guidance tokens $\varnothing_{1:T}^\star$

---

3: $\quad \mathbf{x}_T^\omega \leftarrow \mathbf{u}, \varnothing_T^\star \leftarrow \varnothing$
4: **for** $t = T, ..., 1$ **do**
5: $\quad \mathfrak{C}' \leftarrow \{\mathbf{c}_{\text{sem}}, \varnothing_t^\star, \theta\}$
6: $\quad \mathbf{x}_{t-1}^\omega \leftarrow h_{t|\mathfrak{C}'}^\omega(\mathbf{x}_t^\omega)$
7: $\quad \varnothing_t^\star \leftarrow \text{CTA}(\varnothing_t^\star, \mathbf{x}_{t-1}, \mathbf{x}_{t-1}^\omega)$
8: $\quad \varnothing_{t-1}^\star \leftarrow \varnothing_t^\star$
9: **end for**
10: **Return** $\varnothing_{1:T}^\star$

---

# C. Extended Related Work

**Controllable Representation Learning.** Our work contributes to integrating Pearlian causality (Pearl, 2009) into unsupervised disentangled representation learning (Higgins et al., 2017; Burgess et al., 2018; Kim & Mnih, 2018; Chen et al., 2018; Kumar et al., 2018; Peebles et al., 2020). These approaches aim to learn semantically meaningful, uncorrelated latent factors using modified VAEs (Kingma, 2013), which can aid with controllable generation. However, Locatello et al. (2019; 2020) demonstrate that unsupervised disentanglement is impossible without inductive biases in both models and datasets. To address this, our mechanisms operate within an SCM, with control over the data generation process when possible. Recent work, builds diffusion models with controllable representations (Batzolis et al., 2023; Pandey et al., 2022; Yang et al., 2023b; Mittal et al., 2023; Zhang et al., 2022). Some works introduce semantics into diffusion by interpreting low-rank projections of intermediate images (Park et al., 2023; Haas et al., 2024; Wang et al., 2024), or by employing diffusion models as decoders within VAEs (Preechakul et al., 2022; Pandey et al., 2022; Zhang et al., 2022; Batzolis et al., 2023; Abstreiter et al., 2021), where encoders capture controllable semantics. These approaches improve perceptual quality and identity preservation in image editing, inspiring our semantic mechanisms. Additional constraints on diffusion autoencoding frameworks (Hwa et al., 2024; Cho et al., 2023) further enhance interpretability and fairness. Explicit control, essential for structured counterfactual reasoning, can be attained via guidance using discriminative score functions, either amortised (Ho & Salimans, 2022; Karras et al., 2024; Dinh et al., 2023) or trained independently (Dhariwal & Nichol, 2021).

**Counterfactuals Image Generation.** Generalising Pearl's Causal Hierarchy (Pearl, 2009; Peters et al., 2017; Bareinboim et al., 2022) to high-dimensional data like images poses significant challenges (Zečević et al., 2022; Poinsot et al., 2024). Methods for generating image counterfactuals typically follow the Pearlian method (abduction-action-prediction) (Pawlowski et al., 2020; De Sousa Ribeiro et al., 2023; Xia et al., 2023; Wu et al., 2024; Pan & Bareinboim, 2024; Dash et al., 2022), relying on deep SCMs (DSCMs) where neural networks trained on observational data implement mechanisms. These DSCMs have used models such as normalizing flows (Papamakarios et al., 2021; Winkler et al., 2019), VAEs (Kingma, 2013; Pandey et al., 2022; Sohn et al., 2015), GANs (Goodfellow et al., 2014; Mirza & Osindero, 2014), and HVAEs (Vahdat & Kautz, 2020; Child, 2020) to generate images. DSCMs have also reduced bias counterfactual image edits (Xia et al., 2024), perform data imputations (Ibrahim et al., 2024), and several medical imaging scenarios (Ravi et al., 2019; Reinhold et al., 2021; Rasal et al., 2022). Some approaches generate counterfactuals without explicitly performing abduction (Sauer & Geiger, 2021). For example, (Shen et al., 2022; Yang et al., 2021) incorporate an SCM prior into the latent space of a VAE, but these methods can be difficult to train and scale. Some approaches avoid explicit abduction (Sauer & Geiger, 2021), while others embed an SCM prior into a VAE's latent space (Shen et al., 2022; Yang et al., 2021) which are difficult to train and scale. Methods focusing solely on interventional distributions (Kocaoglu et al., 2017; Rahman et al., 2024) cannot generate counterfactuals. Many studies loosely use the term "counterfactual" to describe structured image perturbations aimed at explaining or interpreting model behaviour (Van Looveren & Klaise, 2021; Fang et al., 2024; Shen et al., 2024; Schut et al., 2021; Kladny et al., 2023; Taylor-Melanson et al., 2024; Augustin et al., 2022; Atad et al., 2024). Recent diffusion-based counterfactual approaches include (Sanchez & Tsaftaris, 2022), whose method diverges from Pearlian abduction and is not demonstrated on large parent sets with complex causal relationships. Potentially, our closest work is (Komanduri et al., 2024) which extends (Preechakul et al., 2022) who implicitly learn their SCM on latent variables given a causal graph in the style of Yang et al. (2021). They, however, use the aggregate diffusion posterior for spatial abduction, followed by classifier-free guidance; we instead present an independent DSCM mechanism for stability, scalability and identity presentation on complex, high-resolution datasets and explicit control over parent sets. We leave identifiability considerations for future work (Hyvarinen & Morioka, 2016; Hyvarinen et al., 2019; Khemakhem et al., 2020a; Li et al., 2019; Sorrenson et al., 2020; Khemakhem et al., 2020b; Roeder et al., 2021; Willetts & Paige, 2021).

**Semantic Image Editing.** Latent diffusion models (Rombach et al., 2022; Podell et al., 2023; Saharia et al., 2022; Nichol et al., 2021) have become standard for creative text-driven image editing (Wang et al., 2024) and image-to-image translation tasks (Meng et al., 2021; Yang et al., 2023a), often using structured analysis and manipulation of cross-attention maps (Hertz et al., 2022; Tang et al., 2022; Epstein et al., 2023; Tumanyan et al., 2023; Sanchez et al., 2022a; Tian et al., 2023). Some methods fine-tune entire models for small datasets (Ruiz et al., 2023; Wang et al., 2023b; Gal et al., 2022) or use masking to localise edits and avoid spurious correlations (Couairon et al., 2022; Pérez-García et al., 2023). We focus on test-time optimisations, using objectives that align inverse and generative trajectories to enhance identity preservation by tuning guidance tokens (Mokady et al., 2023; Wang et al., 2023a; Tang et al., 2024), for their relative efficiency and robustness without masking. SCMs have been applied to text-guided diffusion editing (Zečević et al., 2022; Song et al., 2024; Gu et al., 2023; Prabhu et al., 2023), but text control often lacks the precision for counterfactual reasoning in real-world scenarios.

# D. Architectures

## D.1. Diffusion Mechanisms

We modify the architecture in DiffAE to be trained for conditioning and classifier-free guidance, in which projections of the conditions are added to the timestep embedding. We also use EMA on model parameters at every training step. Additionally, we modify the encoder to include a mean and log-variance, which we reparameterise during training with Equation (13). All images are normalised between $[-1, 1]$. In the case of Morpho-MNIST datasets, the digit class ($d$) is one-hot encoded. For CelebA and CelebA-HQ, we use `torchvision.transforms.RandomHorizontalFlip(p=0.5)` for pre-processing. For EMBED, we use the preprocessing used by (Schueppert et al., 2024) to train their classifiers. We outline the architectures and training procedures of our semantic mechanisms in Table 4; spatial mechanisms are implemented without the semantic encoder components.

*Table 4.* Network architecture of our semantic mechanisms.

| PARAMETER | MORPHO-MNIST | CMORPHO-MNIST | CELEBA | CELEBA-HQ | EMBED |
|---|---|---|---|---|---|
| TRAINING SET | 50000 | 50000 | 162770 | 24000 | 13207 |
| VALIDATION SET | 10000 | 10000 | 19867 | 3000 | 3300 |
| TEST SET | 10000 | 10000 | 19962 | 3000 | 5503 |
| RESOLUTION | $28 \times 28 \times 1$ | $28 \times 28 \times 3$ | $64 \times 64 \times 3$ | $64 \times 64 \times 3$ | $192 \times 192 \times 3$ |
| BATCH SIZE | 128 | 128 | 128 | 128 | 32 |
| EPOCHS | 1000 | 1000 | 6000 | 6000 | 1000 |
| BASE CHANNELS | 64 | 64 | 64 | 64 | 96 |
| ATTENTION RESOLUTION | - | - | [16] | [16] | - |
| CHANNEL MULTIPLIERS | [1,4,8] | [1,4,8] | [1,2,4,8] | [1,2,4,8] | [1,1,2,2,4,4] |
| RESNET BLOCKS | 1 | 1 | 2 | 2 | 2 |
| RESNET DROPOUT | - | - | 0.1 | 0.1 | 0.1 |
| SEM. ENCODER BASE CH. | 64 | 64 | 32 | 32 | 96 |
| SEM. ENC. ATTN. RESOLUTION | - | - | [16] | [16] | - |
| SEM. ENC. CH. MULT. | [1,2,4,8] | [1,2,4,8] | [1,2,4,8,8] | [1,2,4,8,8] | [1,1,2,2,4,4,4] |
| SEM. ENC. RESNET BLOCKS | 1 | 1 | 2 | 2 | 2 |
| SEM. ENC. RESNET DROPOUT | 0.1 | 0.1 | 0.1 | 0.1 | 0.1 |
| **z** SIZE | 8 | 8 | 32 | 32 | 512 |
| NUM. **pa** VARIABLES | 4 | 4 | 7 | 7 | 4 |
| **pa** SIZE | 13 | 13 | 7 | 7 | 4 |
| NOISE SCHEDULER | LINEAR | LINEAR | LINEAR | LINEAR | COSINE |
| LEARNING RATE | 1e-4 | | | | |
| OPTIMISER | ADAM (NO WEIGHT DECAY) | | | | |
| EMA DECAY FACTOR | 0.9999 | | | | |
| TRAINING $T$ | 1000 | | | | |
| DIFFUSION LOSS | MSE WITH NOISE PREDICTION | | | | |

## D.2. Effectiveness Classifiers

**Morpho-MNIST.** We evaluate the effectiveness of the digit class $d$ using the following simple CNN trained for 100 epochs with learning rate $1e - 3$ and batch size 256, achieving $\approx 99.5\%$ accuracy:

```python
from torch import nn

class Classifier(nn.Module):
    super().__init__()
    self.model = nn.Sequential(
        nn.Flatten(),
        nn.Linear(28 * 28, 128),
        nn.ReLU(),
```

```
        nn.Linear(128, 64),
        nn.ReLU(),
        nn.Linear(64, 10),
    )

    def forward(self, x):
        return self.model(x)
```

**CelebA-HQ.** We evaluate the effectiveness of *Eyeglasses* and *Smiling* using a classifier with a ResNet backbone:

```
from torchvision.models import resnet50
from torch import nn

class Classifier(nn.Module):

    def __init__(self):
        super().__init__()
        self.model = resnet50(weights=ResNet50_Weights.IMAGENET1K_V2)
        self.model.fc = nn.Sequential(
            nn.Linear(self.model.fc.in_features, 1024),
            nn.ReLU(),
            nn.Dropout(0.25),
            nn.Linear(1024, 1),
        )

    def forward(self, x):
        return self.model(x)
```

We train this classifier for 100 epochs with a batch size of 128 using the Adam (Kingma, 2014) optimiser, with a starting learning rate of $1e-3$, $\beta_1 = 0.9$, $\beta_2 = 0.999$ and weight decay 0.01. To improve regularisation we use `torchvision.transforms.RandomHorizontalFlip(p=0.5)` for preprocessing. To address the imbalance in *Eyeglasses* and *Smiling*, we also use a weighted random sampler with replacement. To stabilise training, we use EMA on the model parameters with a decay rate 0.999. Both classifiers achieve $\approx 97\%$ accuracy.

**EMBED.** We use downscale the multi-label classifier used in (Schueppert et al., 2024) for skin and circular marker detection for $192 \times 192$ mammograms, achieving a ROC-AUC of .91 on the test set.

# E. Morpho-MNIST

## E.1. Dataset Details

We construct the following SCM using the Morpho-MNIST library (Castro et al., 2019), to extend the casual modelling scenarios in (Pawlowski et al., 2020; De Sousa Ribeiro et al., 2023). Mechanisms are defined as follows:

$$d := f_d(\boldsymbol{\epsilon}_d) = \boldsymbol{\epsilon}_d, \qquad\qquad \boldsymbol{\epsilon}_d \sim \text{MNIST} \tag{31}$$
$$s := f_s(d, \boldsymbol{\epsilon}_s) = -27 + d \cdot 6 + 3 \cdot \boldsymbol{\epsilon}_s \qquad\qquad \boldsymbol{\epsilon}_s \sim \mathcal{N}(0,1) \tag{32}$$
$$t := f_t(\boldsymbol{\epsilon}_t) = 0.5 + \boldsymbol{\epsilon}_t, \qquad\qquad \boldsymbol{\epsilon}_t \sim \Gamma(10,5) \tag{33}$$
$$i := f_i(t, \boldsymbol{\epsilon}_i) = 191 \cdot \sigma(0.5 \cdot \boldsymbol{\epsilon}_i + 2t - 5), \qquad\qquad \boldsymbol{\epsilon}_i \sim \mathcal{N}(0,1) \tag{34}$$
$$\mathbf{x} := f_{\mathbf{x}}(i, t, d, s, \boldsymbol{\epsilon}_{\mathbf{x}}) = \text{Set}_i(i, d, \text{Set}_t(t, d, \text{Set}_s(s, d, \boldsymbol{\epsilon}_{\mathbf{x}}))), \qquad\qquad \boldsymbol{\epsilon}_{\mathbf{x}} \sim \text{MNIST} \tag{35}$$

where $\text{Set}_i(\cdot)$, $\text{Set}_t(\cdot)$ and $\text{Set}_s(\cdot)$ are functions provided by Morpho-MNIST that change the intensity $i$, thickness $t$ and slant $s$ of an image in the original MNIST dataset $\boldsymbol{\epsilon}_{\mathbf{x}}$. We use the true mechanisms $f_i(\cdot)$, $f_t(\cdot)$, $f_s(\cdot)$ and $f_d(\cdot)$ to implement our DSCM (Figure 2a), with the image generating mechanism implemented using our diffusion-based formulations. When using these mechanisms to generate our dataset, we ensure that all digits have full support of the slant:

$$s := \begin{cases} f_s(d, \boldsymbol{\epsilon}_s), & \text{if } b = 0, \text{where } b \sim \text{Bern}(0.2) \text{ and } \boldsymbol{\epsilon}_s \sim \mathcal{N}(0,1), \\ \boldsymbol{\epsilon}_s, & \text{Otherwise, where } \boldsymbol{\epsilon}_s \sim \text{MNIST}. \end{cases} \tag{36}$$

Our dataset follows the original MNIST dataset splits.

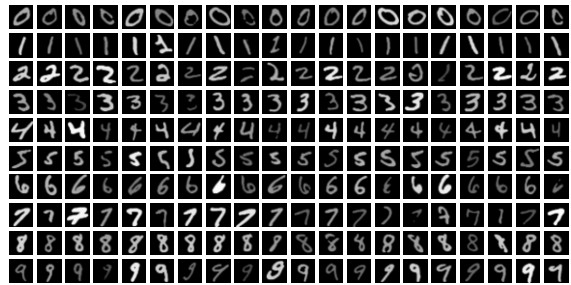

*Figure 7.* Examples from Morpho-MNIST

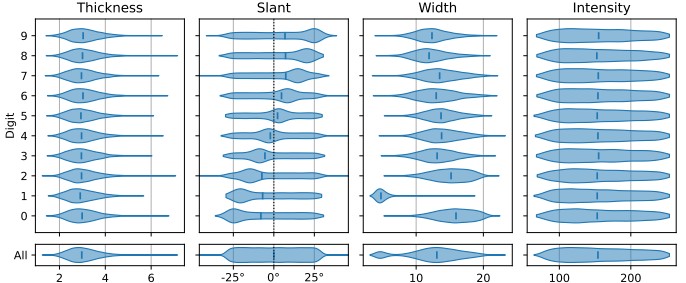

*Figure 8.* Features distributions in Morpho-MNIST

## E.2. Extra Results

*Table 5.* Soundness of Morpho-MNIST image counterfactuals under $do(t)$ and $do(i)$ using DSCMs modelling the SCM in Appendix E. Effectiveness for digit class ($d$) is measured using accuracy (Acc) from a pre-trained classifier and mean absolute percentage error (MAPE) for slant ($s$), thickness ($t$) and intensity ($i$). Counterfactuals are normalised to $[0, 1]$ to measure composition (COMP.) and reversibility (REV.). All metrics are scaled by $\times 10^{-2}$, except for MAPE ($s$), which is scaled by $\times 10^{-1}$, and Acc ($d$), which remains unscaled.

| | THICKNESS INTERVENTION ($do(t)$) | | | | | INTENSITY INTERVENTION ($do(i)$) | | | | |
|---|---|---|---|---|---|---|---|---|---|---|
| | EFFECTIVENESS | | | | REV. | EFFECTIVENESS | | | | REV. |
| MECHANISM | MAPE ($t$) ↓ | MAPE ($i$) ↓ | MAPE ($s$) ↓ | Acc ($d$) ↑ | $L_1$ ↓ | MAPE ($t$) ↓ | MAPE ($i$) ↓ | MAPE ($s$) ↓ | Acc ($d$) ↑ | $L_1$ ↓ |
| VAE (Pawlowski et al., 2020) | 4.48 | 6.76 | 3.88 | 97.75 | 4.26 | 8.06 | 7.55 | 4.83 | 97.85 | 4.31 |
| HVAE (De Sousa Ribeiro et al., 2023) | 3.05 | 0.675 | 1.82 | 95.61 | 0.678 | 3.92 | 0.471 | 1.60 | 94.92 | 0.580 |
| VCI (Wu et al., 2024) | 3.08 | 0.626 | 0.913 | 92.97 | 2.10 | 6.99 | 2.61 | 3.97 | 82.52 | 3.75 |
| SPATIAL: | 2.99 | 0.506 | 1.75 | 96.55 | 1.34 | 4.33 | 0.735 | 3.51 | 92.39 | 1.35 |
| $\{\omega = 1.5, p_{\varnothing} = 0.1\}$ | 1.96 | 0.355 | 1.44 | 99.61 | 2.35 | 3.46 | 0.496 | 3.07 | 97.17 | 2.28 |
| $\{\omega = 3, p_{\varnothing} = 0.1\}$ | 1.95 | 0.296 | 1.16 | 99.89 | 4.78 | 3.04 | 0.629 | 2.10 | 95.84 | 4.88 |
| $\{\omega = 4.5, p_{\varnothing} = 0.1\}$ | 1.98 | 0.327 | 1.26 | 99.98 | 5.70 | 2.43 | 0.936 | 1.91 | 98.63 | 5.70 |
| $\{\omega = 1.5, p_{\varnothing} = 0.5\}$ | 2.29 | 0.478 | 2.78 | 98.93 | 1.48 | 2.91 | 0.637 | 2.92 | 97.75 | 1.07 |
| $\{\omega = 3, p_{\varnothing} = 0.5\}$ | 2.21 | 0.389 | 1.02 | 99.71 | 3.10 | 2.93 | 1.13 | 1.34 | 98.35 | 3.45 |
| $\{\omega = 4.5, p_{\varnothing} = 0.5\}$ | 2.03 | 0.382 | 2.15 | 99.90 | 3.37 | 2.56 | 1.02 | 1.59 | 99.12 | 4.78 |
| SEMANTIC: | 4.17 | 0.718 | 3.18 | 94.43 | 1.72 | 5.90 | 1.62 | 3.00 | 87.50 | 2.23 |
| $\{\omega = 1.5, p_{\varnothing} = 0.1\}$ | 2.36 | 1.29 | 2.30 | 98.54 | 1.86 | 3.38 | 1.40 | 2.12 | 96.39 | 1.70 |
| $\{\omega = 3, p_{\varnothing} = 0.1\}$ | 2.16 | 1.01 | 1.56 | 99.51 | 3.52 | 3.41 | 1.45 | 2.11 | 97.75 | 3.27 |
| $\{\omega = 4.5, p_{\varnothing} = 0.1\}$ | 2.24 | 0.757 | 2.73 | 99.61 | 4.72 | 4.21 | 1.55 | 2.56 | 98.24 | 4.43 |

*Table 7.* Soundness of Morpho-MNIST image counterfactuals generated under $do(d)$ from DiffSCM modelling the relationship $d \rightarrow \mathbf{x}$, in which the digit class ($d$) is the only parent of the image ($\mathbf{x}$), with data generated from the true SCM in Appendix E. Effectiveness (EFF.) is measured by the accuracy (Acc) of a pre-trained classifier. Counterfactuals are normalised to $[0, 1]$ to measure composition (COMP.) and reversibility (REV.).

| MECHANISM | COMP. $L_1 \downarrow (\times 10^{-2})$ | EFF. Acc $\uparrow$ | REV. $L_1 \downarrow (\times 10^{-2})$ |
|---|---|---|---|
| DIFFSCM (ORIGINAL) | 0.410 | 17.02 | 1.31 |
| DIFFSCM (OURS) $\{\omega = 10\}$ | 0.605 | 73.83 | 3.92 |
| DIFFSCM (OURS) $\{\omega = 20\}$ | 0.945 | 96.00 | 3.94 |
| DIFFSCM (OURS) $\{\omega = 30\}$ | 1.22 | 98.73 | 4.42 |

*Table 6.* Soundness of Morpho-MNIST image counterfactuals with mechanisms conditioned on intensity ($i$) and thickness ($t$) from the dataset in Appendix E.1. Effectiveness for thickness ($t$) and intensity ($i$) is measured using mean absolute percentage error (MAPE). Here, we use simulated interventions $do(i)$ and $do(t) = do(i, t)$. Counterfactuals are normalised to $[0, 1]$ to measure composition (COMP.) and reversibility (REV.). All metrics are scaled by $\times 10^{-2}$.

| | THICKNESS INTERVENTION ($do(t)$) | | | INTENSITY INTERVENTION ($do(i)$) | | | NULL INT. ($do(\mathbf{pa})$) |
|---|---|---|---|---|---|---|---|
| | EFFECTIVENESS | | REV. | EFFECTIVENESS | | REV. | COMP. |
| MECHANISM | MAPE ($t$) $\downarrow$ | MAPE ($i$) $\downarrow$ | $L_1 \downarrow$ | MAPE ($t$) $\downarrow$ | MAPE ($i$) $\downarrow$ | $L_1 \downarrow$ | $L_1 \downarrow$ |
| VCI (Wu et al., 2024) | 5.09 | 0.949 | 1.17 | 9.58 | 5.89 | 2.74 | 0.665 |
| SPATIAL: | 3.45 | 0.631 | 1.595 | 5.52 | 1.07 | 1.765 | 0.0463 |
| $\{\omega{=}1.5,\ p_\varnothing{=}0.1\}$ | 2.44 | 0.381 | 2.075 | 4.42 | 0.737 | 2.49 | 0.3905 |
| $\{\omega{=}3,\ p_\varnothing{=}0.1\}$ | 2.15 | 0.357 | 3.56 | 3.88 | 0.615 | 4.645 | 1.10 |

### E.3. Improving DiffSCM

The results in Table 7 demonstrate that DiffSCM's original selection of guidance scale, based on their proposed counterfactual latent divergence metric, substantially limits effectiveness. By exploring higher guidance scales beyond those originally reported and training the unconditional diffusion model for 500K steps, compared to the 30K used in the original work, we are able to improve effectiveness significantly. However, this gain in effectiveness comes at the expense of identity preservation, as reflected by the increasing composition and reversibility errors. As $\omega$ grows, the influence of the unconditional diffusion model diminishes relative to the anti-causal classifier, meaning counterfactuals increasingly satisfy the desired class intervention while deviating from the spatial exogenous noise characteristics of the original observation. Comparing ablations at $\omega = 20$ and $\omega = 30$, where effectiveness becomes comparable to our mechanisms in Table 1, we find that DiffSCM's composition is similar to our spatial mechanisms but does not outperform our semantic mechanisms, while reversibility remains substantially worse compared to our methods. We hypothesise that learning an amortised anti-causal generative classifier for guidance, as in our method, is more advantageous for reversibility than using a separately trained classifier.

## E.4. Morpho-MNIST Counterfactuals

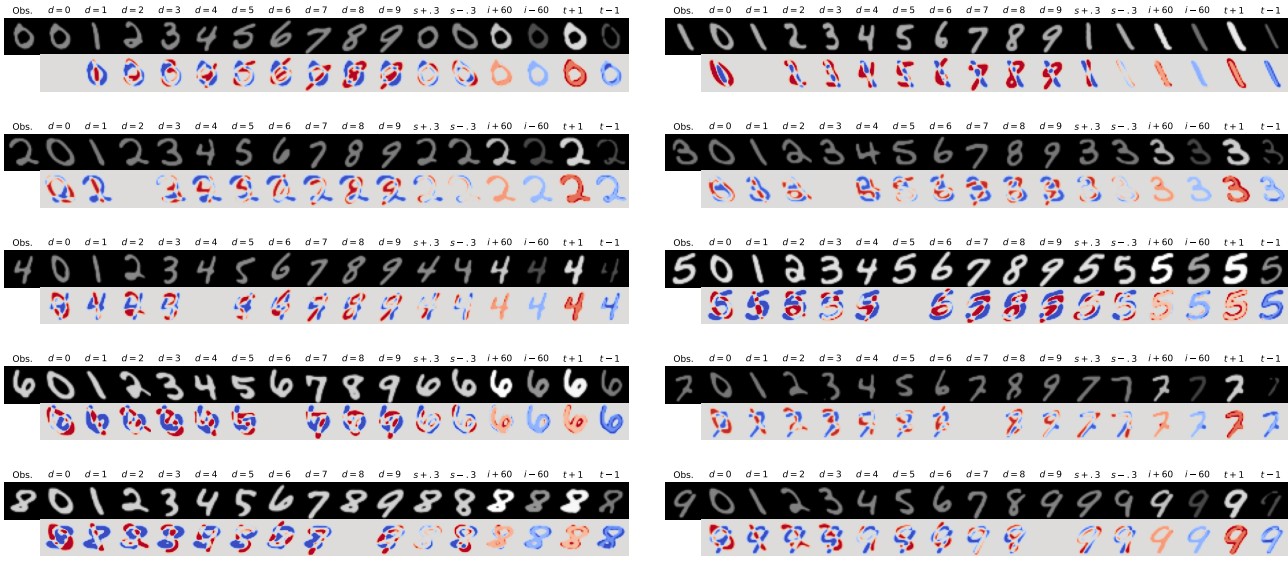

*Figure 9.* Morpho-MNIST ($28 \times 28$) counterfactuals generated using an amortised, anti-causally guided semantic mechanism ($p_\varnothing = 0.1, \omega = 1.5$) based on the DSCM shown in Figure 2a. Interventions are shown above the top row and the bottom row visualises total causal effects (red: increase, blue: decrease), refer to (Appendix A.2) for details.

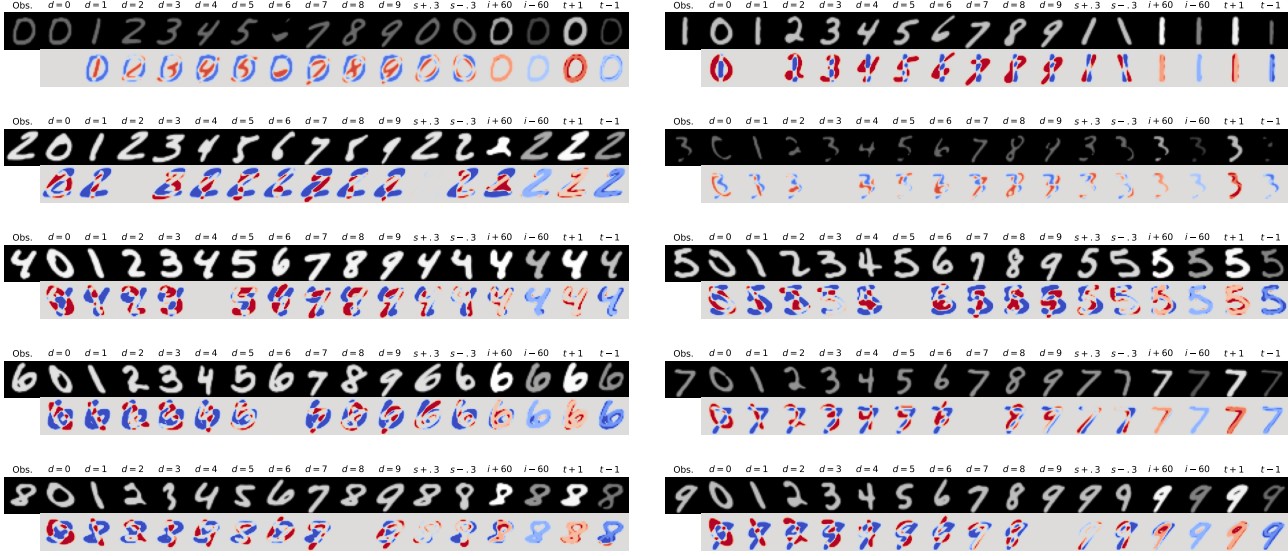

*Figure 10.* Morpho-MNIST ($28 \times 28$) counterfactuals generated using an amortised, anti-causally guided spatial mechanism ($p_\varnothing = 0.1, \omega = 1.5$) based on the DSCM shown in Figure 2a. Interventions are shown above the top row and the bottom row visualises total causal effects (red: increase, blue: decrease), refer to (Appendix A.2) for details.

# F. Colourised Morpho-MNIST

## F.1. Dataset Details

We construct a colourised variant of the dataset in Appendix E.1 by using the Morpho-MNIST library to implement the following structural causal model, which extends the causal modelling scenario in (Monteiro et al., 2023):

$$d := f_d(\boldsymbol{\epsilon}_d), \qquad\qquad \boldsymbol{\epsilon}_d \sim \text{MNIST} \qquad (37)$$

$$t := f_t(\boldsymbol{\epsilon}_t) = 0.5 + \boldsymbol{\epsilon}_t, \qquad\qquad \boldsymbol{\epsilon}_t \sim \Gamma(10, 5) \qquad (38)$$

$$s := f_s(d, \boldsymbol{\epsilon}_s) = -27 + d \cdot 6 + 3 \cdot \boldsymbol{\epsilon}_s \qquad\qquad \boldsymbol{\epsilon}_s \sim \mathcal{N}(0, 1) \qquad (39)$$

$$h := f_h(d, \boldsymbol{\epsilon}_h) = 0.1 \cdot d + 0.05 + 0.05 \cdot \boldsymbol{\epsilon}_h \qquad\qquad \boldsymbol{\epsilon}_h \sim \mathcal{N}(0, 1) \qquad (40)$$

$$\mathbf{x} := f_{\mathbf{x}}(h, t, d, s, \boldsymbol{\epsilon}_{\mathbf{x}}) = \text{Set}_h(h, d, \text{Set}_t(t, d, \text{Set}_s(s, d, \boldsymbol{\epsilon}_{\mathbf{x}}))), \qquad\qquad \boldsymbol{\epsilon}_{\mathbf{x}} \sim \text{MNIST}, \qquad (41)$$

where $\text{Set}_t(\cdot)$ and $\text{Set}_s(\cdot)$ are functions in Morpho-MNIST that set the thickness $t$ and slant $s$ of an image, and we implement $\text{Set}_h(\cdot)$ to set its hue $h$. We use the true mechanisms $f_h(\cdot)$, $f_t(\cdot)$, $f_s(\cdot)$ and $f_d(\cdot)$ to implement our DSCM (Figure 13a), with the image generating mechanism implemented using our diffusion-based formulations. When using these mechanisms to generate our dataset, we ensure that all digits have full support of the slant and hue:

$$s := \begin{cases} f_s(d, \boldsymbol{\epsilon}_s), & \text{if } b = 0, \text{where } b \sim \text{Bern}(0.2) \text{ and } \boldsymbol{\epsilon}_s \sim \mathcal{N}(0, 1), \\ \boldsymbol{\epsilon}_s, & \text{Otherwise, where } \boldsymbol{\epsilon}_s \sim \text{MNIST}, \end{cases} \qquad (42)$$

$$h := \begin{cases} f_h(d, \boldsymbol{\epsilon}_h), & \text{if } b = 0, \text{where } b \sim \text{Bern}(0.5) \text{ and } \boldsymbol{\epsilon}_h \sim \mathcal{N}(0, 1), \\ \boldsymbol{\epsilon}_h, & \text{Otherwise, where } \boldsymbol{\epsilon}_h \sim \mathcal{U}[0, 1]. \end{cases} \qquad (43)$$

Our dataset follows the original MNIST dataset splits.

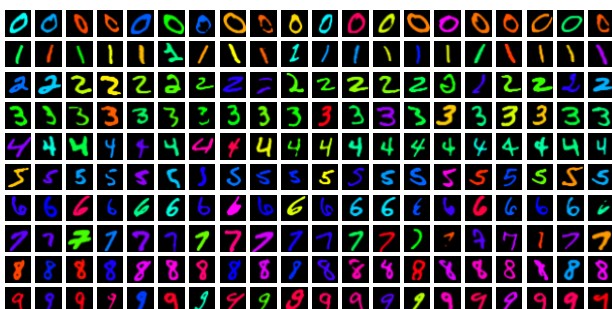

*Figure 11.* Examples from colour Morpho-MNIST

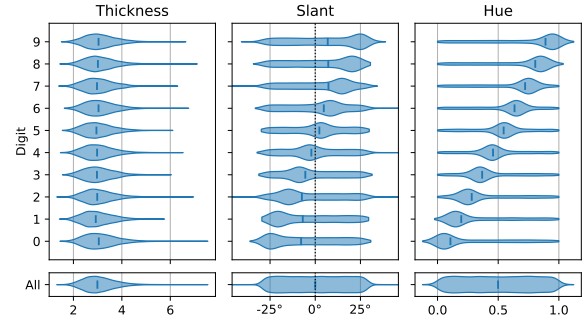

*Figure 12.* Features distributions in colour Morpho-MNIST

## F.2. Qualitative Results

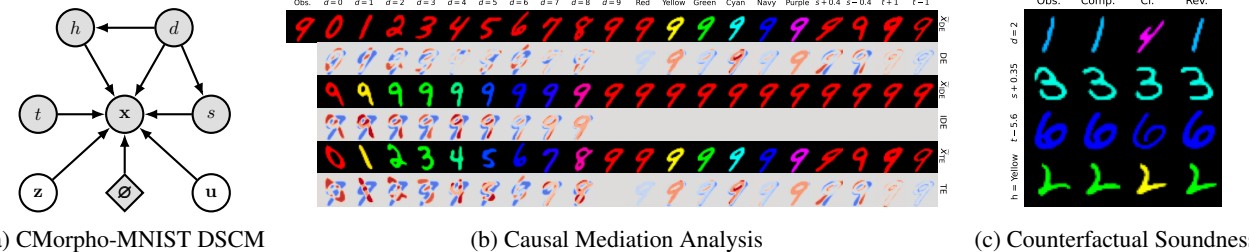

(a) CMorpho-MNIST DSCM        (b) Causal Mediation Analysis        (c) Counterfactual Soundness

*Figure 13.* Colourised Morpho-MNIST ($28 \times 28$) counterfactuals generated using an amortised, anti-causally guided semantic mechanism ($p_\varnothing = 0.1$, $\omega = 1.5$). (a) depicts the DSCM: $h$ is hue, $d$ is digit class, $s$ is slant, $t$ is thickness and $\mathbf{x}$ is the image. (b) depicts image counterfactuals and causal mediation analysis (Appendix A.2): interventions are shown above the top row and the bottom row visualises total causal effects (red: increase, blue: decrease). (c) illustrates counterfactual soundness (Obs: Observation, Comp: Composition, Cf: Counterfactual, Rev: Reversibility).

## G. CelebA

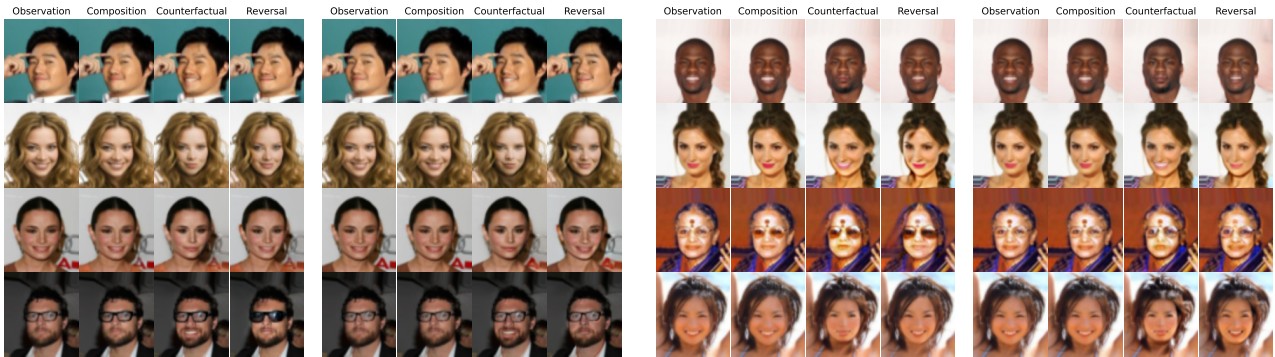

(a) Guided spatial mechanism (left) with dynamic abduction (right).  (b) Guided semantic mech. (left) with dynamic abduction (right).

*Figure 14.* CelebA ($64 \times 64$) counterfactuals generated using our amortised, anti-causally guided mechanisms in the DSCM in Figure 3a.

## H. CelebA-HQ

### H.1. Additional Results

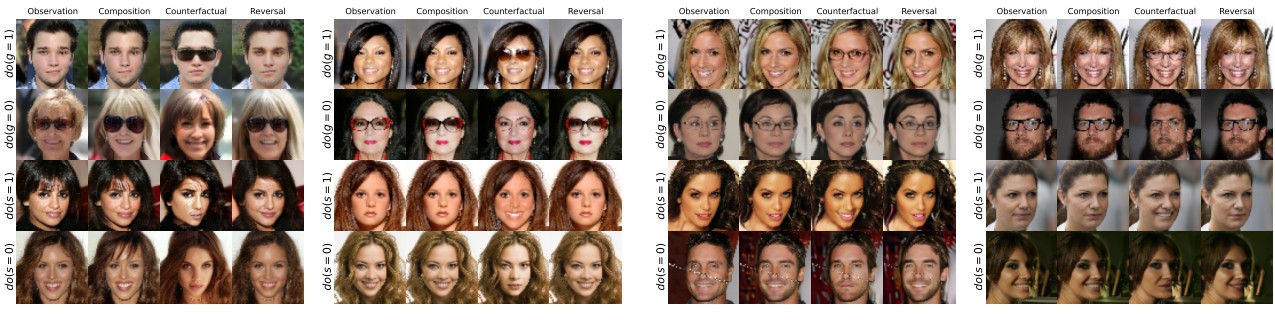

(a) Guided spatial mech. with $p_\varnothing = 0.1$ (left) and $p_\varnothing = 0.5$ (right).  (b) Guided sem mech. with $p_\varnothing = 0.1$ (left) and $p_\varnothing = 0.5$ (right).

*Figure 15.* CelebA-HQ ($64 \times 64$) counterfactuals using our amortised, anti-causally guided mechanisms in the DSCM in Figure 3a with $\omega = 2$. Notice that identity preservation improves in each figure from left to right.

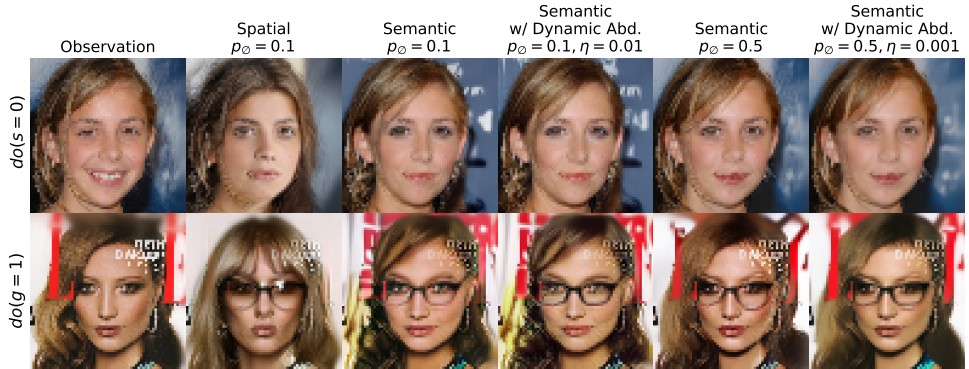

*Figure 16.* CelebA-HQ ($64 \times 64$) counterfactuals generated using amortised, anti-causally guided mechanisms. Notice that identity preservation improves as we use increasingly complex semantic abduction procedures and increase the value of $p_\varnothing$.

## H.2. Baselines

We refer readers to Figures 2e and 2g in (Monteiro et al., 2023) for an empirical comparison of face image counterfactuals under $do(s)$ and $do(g)$ interventions. Their results exhibit notable background changes and reduced image fidelity, issues that our dynamic abduction method largely corrects. Additionally, their Table 6 reports effectiveness levels comparable to ours, displaying a similar trade-off between effectiveness and identity preservation, in this case depending on the number of hierarchical latent variables abducted. This model forms the backbone of the HVAE used by (De Sousa Ribeiro et al., 2023), and was subsequently adopted by (Melistas et al., 2024) for causal modelling of face images on CelebA (Liu et al., 2015). To implement the computational graph in Figure 3a for CelebA-HQ, we train the model provided by (Melistas et al., 2024). However, we observe that without extensive hyperparameter tuning, particularly when handling many confounders, and without post-hoc counterfactual fine-tuning, the model exhibits very poor effectiveness. VCI (Wu et al., 2024) introduces a counterfactual regularisation term and an adversarial loss into the autoencoding objective. They extend the models from (De Sousa Ribeiro et al., 2023; Monteiro et al., 2023) to generate face counterfactuals on the CelebA-HQ subset of images taken directly from CelebA. As such, VCI applies center-cropping to all images following the preprocessing procedure of DEAR (Shen et al., 2022). In contrast, when we train VCI directly on CelebA-HQ and without center-cropping, to ensure compatibility with our pre-trained classifiers, we observe a significant drop in counterfactual fidelity. Table 8 provides metrics for the failure cases of the baselines we have discussed above. We acknowledge that with additional hyperparameter tuning and alternative evaluation setups, i.e. using random-cropping, these models may perform better. However, in comparison, our diffusion-based mechanisms with sufficient model parameters achieve strong performance without the need for center-cropping, adversarial/counterfactual losses or extensive hyperparameter tuning. We anticipate that future work leveraging high-resolution images, where diffusion models are known to excel, and adopting an LDM-style architecture will enable scaling to even higher resolutions and further improve identity preservation.

*Table 8.* Soundness of CelebA-HQ image counterfactuals generated using VCI (Wu et al., 2024) without center-cropping in preprocessing and HVAE (Melistas et al., 2024) without counterfactual fine-tuning. Effectiveness is measured using the F1-scores from pre-trained classifiers for eyeglasses ($g$) and smiling ($s$).

| | EYEGLASSES INTERVENTION ($do(g)$) | | SMILING INTERVENTION ($do(s)$) | |
|---|---|---|---|---|
| MECHANISM | F1 ($s$) ↑ | F1 ($g$) ↑ | F1 ($s$) ↑ | F1 ($g$) ↑ |
| VCI (Wu et al., 2024) | 97.84 | 3.39 | 33.81 | 99.58 |
| HVAE (Melistas et al., 2024) | 90.05 | 65.31 | 75.33 | 95.82 |

# I. EMBED

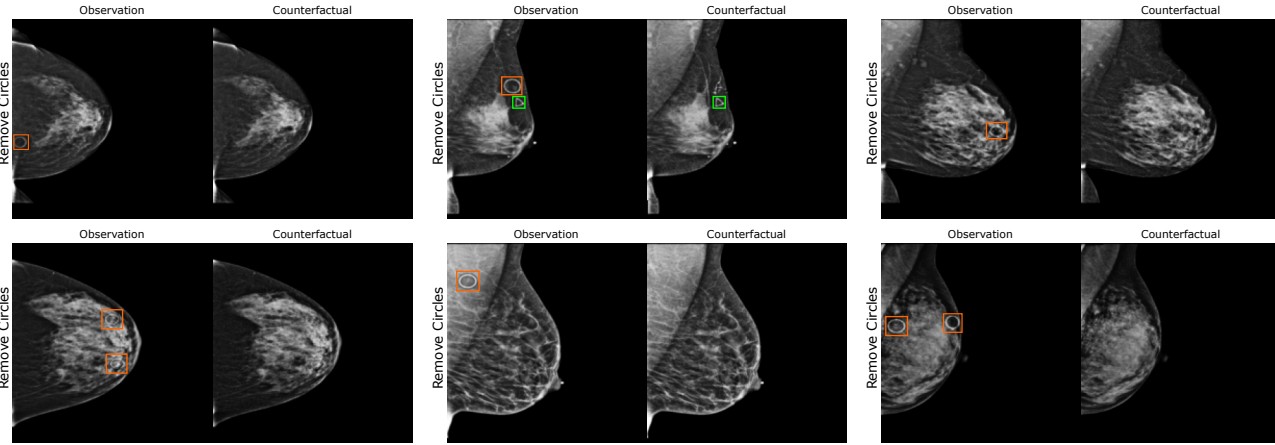

*Figure 17.* EMBED ($192 \times 192$) counterfactuals using an amortised, anti-causally guided semantic mechanism with ($p_\varnothing = 0.1, \omega = 1.2$) for circular skin marker removal. We find that larger $\omega$ improves effectiveness whilst compromising illumination and breast density.

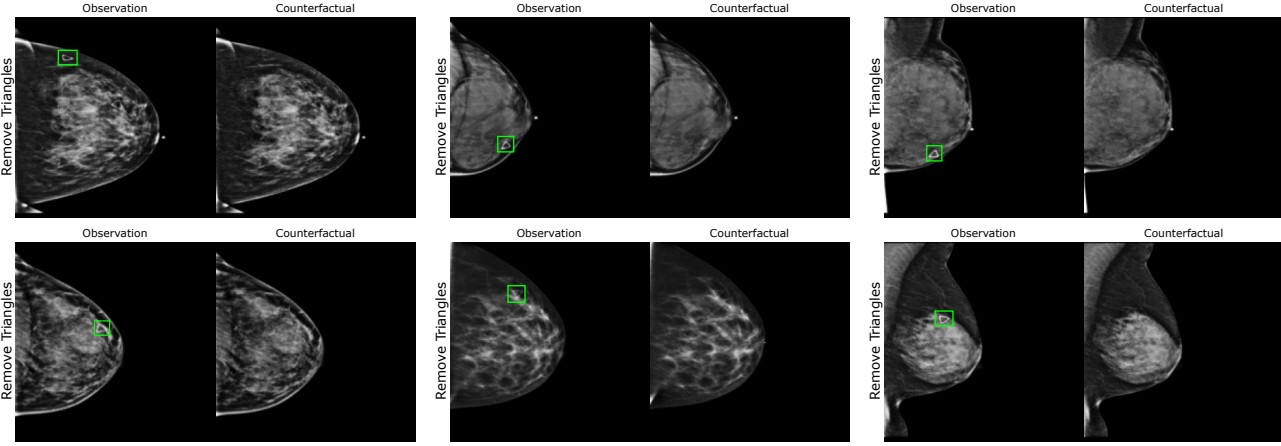

*Figure 18.* EMBED ($192 \times 192$) counterfactuals using an amortised, anti-causally guided semantic mechanism with ($p_\varnothing = 0.1, \omega = 1.2$) for triangular skin marker removal. We find that larger $\omega$ improves effectiveness whilst compromising illumination and breast density.

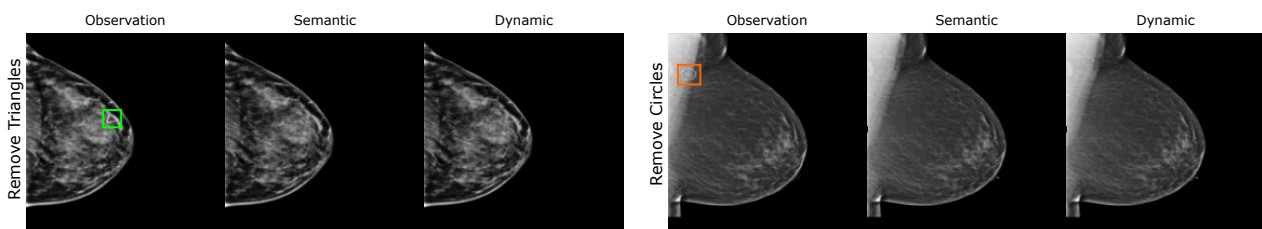

*Figure 19.* EMBED ($192 \times 192$) counterfactuals using an amortised, anti-causally guided semantic mechanism with ($p_\varnothing = 0.1, \omega = 1.2$) with dynamic abduction ($\eta = 0.001$) for triangular and circular marker removal. Counterfactual inference takes $\sim 5$ mins per image. In these cases, dynamic abduction improves the sharpness and brightness of our images.

