# OpenReview forum: "Diffusion Counterfactual Generation with Semantic Abduction"
_ICML.cc/2025/Conference — ICML 2025 poster_

### Official Review · Reviewer_kmep · 2025-02-18

**Overall Recommendation:** 4

**Summary:**

The authors proposed a method that incorporates semantic abduction in diffusion models for the preservation of exogenous noise in counterfactual generations. The inference is conducted with a CFG-style amortised, anti-causally guided DDIM sampling. Results are shown on three datasets: Morpho-MNIST, CelebA-HQ and EMBED.

**Claims And Evidence:**

It's ambiguous. The claim is that this is the first work to consider high-level semantic abduction for diffusion counterfactual. If you strictly consider the meaning of the terminology "semantic abduction" that the authors used, this might be the first, however, I think the proposed method is not novel and has minor contributions among the existing diffusion counterfactual models. Using diffusion models for exogenous abduction is already proposed in DiffSCM [1] and using amortised, anti-causally guided inference is already proposed in DDCM [2], which entails high-level exogenous abduction. Both of these works are not compared in either related work or experiments.

[1] Sanchez, Pedro, and Sotirios A. Tsaftaris. "Diffusion causal models for counterfactual estimation." CLeaR (2022).

[2] Xu, Sihan, et al. "Inversion-Free Image Editing with Language-Guided Diffusion Models." CVPR. 2024.

**Essential References Not Discussed:**

Yes, prior works in diffusion-based counterfactual models [1][2] and state-of-the-art counterfactual models [3] are not discussed in related work or compared in the experiment section. See "Claims And Evidence" and "Methods And Evaluation Criteria" for details.

[1] Sanchez, Pedro, and Sotirios A. Tsaftaris. "Diffusion causal models for counterfactual estimation." CLeaR (2022).

[2] Xu, Sihan, et al. "Inversion-Free Image Editing with Language-Guided Diffusion Models." CVPR. 2024.

[3] Wu, Yulun, Louie McConnell, and Claudia Iriondo. "Counterfactual Generative Modeling with Variational Causal Inference." ICLR (2024)

**Experimental Designs Or Analyses:**

I checked the soundness of experimental design, and I think the evaluation metrics used for MorphoMNIST are problematic. For a dataset where *the counterfactual truth is available*, there is no reason to use composition and reversibility in favor of the error between counterfactual generation and counterfactual truth. I believe the authors are well-aware that composition, effectiveness, and reversibility are metrics proposed to evaluate results on datasets where *counterfactual truth is not available*. For MorphoMNIST, there is no reason to use composition – the MSE between reconstruction and original image, and reversibility – the MSE between cycle reconstruction and original image, in favor of the counterfactual prediction error – the MSE between counterfactual construction and counterfactual image. A naive model that does nothing other than returning the original image can achieve the perfect composition and reversibility. The counterfactual prediction error, on the other hand, is a true counterfactual evaluation metric on layer 3 of Pearl’s 3 layers of causality and should always be preferred when it’s attainable.

## Update After Rebuttal

I have raised the score to a strong acceptance as the authors have put in an abundant amount of efforts in additional experiments. However, this is the one issue that the authors did not give me an adequate answer for, and since this year's ICML does not seem to allow further discussion with the authors after their **Reply Rebuttal Comment** is posted, I am highlighting my intended response to their **Reply Rebuttal Comment** here for readers' information:

---

*Thanks for the detailed response. Great work in rebuttal. I have raised my score to a 4, assuming that the comparison to DiffSCM will actually be in the final version of the paper. I do not think your argument make sense regarding issue 1, and the concern you [linked](https://openreview.net/forum?id=oeDcgVC7Xh&noteId=is9vzvhbQQ) was literally a clarification that composition and reversibility should be used in CelebA, where counterfactual truth is **not** attainable. So I do not see how this is related to the issue I raised here.*

> *MSE/MAE remain meaningful for composition and reversibility because the target is uniquely defined (ie. the observed image)*

*I think this is such a poor argument for using composition and reversibility over counterfactual error. The counterfactual image **is** uniquely defined, it is just not observed for real-world datasets. It is literally a unique target by the definition of counterfactual and the consistency assumption. You can correct me on this if I'm wrong: I don't believe there is any randomness to the counterfactual image generated by morpho-MNIST when intervention is given. Not only is the target uniquely defined, it **is** the target you care about -- ultimately, you don't care about how good your model can do reconstruction or cycle reconstruction, you care about how good your model can do counterfactual generation. The former is an indirect estimate of model's ability in exogenous noise abduction when the direct estimate, i.e. the latter, cannot be evaluated.*

> *These metrics are also sensitive to sharpness and local artefacts. A poor but blurry counterfactual may yield low MSE [1,2,8].*

*This is the flaw of pixel-wise MSE, it has nothing to do with the comparison of counterfactual error vs. composition & reversibility as all of them are pixel-wise MSE. If this is your argument why counterfactual error is flawed, I can use the exact same argument to tell you composition & reversibility are flawed as well.*

*You are contradicting yourself with these two arguments. You said "MSE/MAE remain meaningful for composition and reversibility because the target is uniquely defined (ie. the observed image)" yet you also link works [1,2,8] where MSE is calculated on uniquely defined observed image to argue MSE/MAE is flawed. Just like the reviewer in the concern you linked, you don't have a coherent argument why composition & reversibility should be preferred over counterfactual error.*

*[1] Zhang et al. "The unreasonable effectiveness of deep features as a perceptual metric." CVPR 2018*

*[2] Wang & Bovik, "Mean squared error: Love it or leave it?" IEEE 2009 - Fig 2*

*[8] Wang et al, "Image quality assessment: from error visibility to structural similarity." IEEE 2004*

---

**Methods And Evaluation Criteria:**

No. On Morpho-MNIST, there is no benchmark comparison to prior diffusion-based counterfactual models and no benchmark comparison to state-of-the-art counterfactual models. On CelebA-HQ, there is no benchmark comparison, period. Poor benchmark comparisons had unfortunately been a bad norm of this field up until 2023, but it is not the norm anymore. Not everyone just simply use VAE and HVAE anymore. The readers should very reasonably expect *at least* three benchmark models being compared to this work:

- DiffSCM [1] – as the prior effort in applying diffusion model in counterfactual modeling

- DDCM [2] – as the state-of-the-art diffusion model for image editing that already proposed the CFG-style [4] *amortised, anti-causally guided inference*.

- VCI [3] – as the current state-of-the-art in counterfactual modeling.

I know this sounds like a lot, but unfortunately that just means this work hasn’t done enough. In my judgement, it is not ready for publication. Again, any reader should *very reasonably* expect these three works being compared to the proposed method for the reasons I listed. I noticed that [1] and [3] are cited, but [2] is not even cited. (Note that image editing is essentially the same as image counterfactual, especially for a work that isn't theoretically-driven. On an empirical level, the difference is simply whether you use the term "counterfacutal" and frame it within a causal formulation or not. On observational datasets, there is no difference.) As for the results, the proposed method does not beat HVAE by a noticeable margin (if it beats HVAE at all) according to Table 1, while VCI [3] beat HVAE by a wide margin. So, it is very reasonable for me to assume that this work performs sub-par compared to state-of-the-art without further benchmark comparison.

Since I do not have benchmark comparisons to evaluate the results, I tried to judge the results by empirical comparison. However, the empirical results the authors showed in Figure 3. b) do not show good ability in exogenous noise abduction either – in most of these pictures, the intervention *greatly* changed the person’s hair style and exhibited some denoising ability (which you do not want for counterfactual modeling) – this seems very sub-par compared to DDCM and VCI, and I’m not sure if it even beats the old non-variational and non-diffusion based method [5].

[1] Sanchez, Pedro, and Sotirios A. Tsaftaris. "Diffusion causal models for counterfactual estimation." CLeaR (2022).

[2] Xu, Sihan, et al. "Inversion-Free Image Editing with Language-Guided Diffusion Models." CVPR. 2024.

[3] Wu, Yulun, Louie McConnell, and Claudia Iriondo. "Counterfactual Generative Modeling with Variational Causal Inference." ICLR (2024).

[4] Ho, Jonathan, and Tim Salimans. "Classifier-free diffusion guidance." (2022).

[5] Zhu, Jun-Yan, et al. "Unpaired image-to-image translation using cycle-consistent adversarial networks." ICCV. 2017.

**Other Comments Or Suggestions:**

I think trying to abduct high-level semantics to aid diffusion models is interesting and intuitive, and I do not want to discourage the authors from keeping exploring this direction. But I think you definitely have to show more evidence of how it works better than / different from existing diffusion models to be more convincing.

**Other Strengths And Weaknesses:**

Strengths:

The paper is clear and well written. Figures are clean and illustrative.

Weaknesses:

See "Claims And Evidence", "Methods And Evaluation Criteria", and "Experimental Designs Or Analyses".

**Questions For Authors:**

No further questions beyond the weaknesses I listed. If the authors can address those that'd be great.

**Relation To Broader Scientific Literature:**

As discussed in "Claims And Evidence" and "Methods And Evaluation Criteria", I do not find any evidence of this work exceling prior works in diffusion-based counterfactual models and state-of-the-art counterfactual models.

**Theoretical Claims:**

N/A. No theoretical claims in this paper.

---

> ### Author Rebuttal · Authors · 2025-03-31
>
> We thank the reviewer for their constructive feedback and are encouraged by their comments that semantic abduction is “interesting and intuitive”. Below, we address the major points raised:
>
> > **“Semantic Abduction” and Novelty**
>
> Refer to our response to reviewer vbtG16 for details regarding the novelty and our contributions.
>
> > **Quality of Abducted Exogenous Noise**
>
> We acknowledge the reviewer’s observation regarding Fig 3b. The denoising training procedure may influence the apparent challenges in exogenous noise abduction. However, we note that in Fig 3c and App H - Fig 16, dynamic semantic abduction and $p_\varnothing > 0.1$ improves identity preservation of hairstyle, skin colour, facial structure and backgrounds, which is especially challenging given that we choose not to center crop faces as in existing methods, such as [2,3]. Additionally, we refer to Fig 5 in [2], which shows changes in skin colour, and Fig 2e/2g in [4], which shows changes in background and image fidelity, which the dynamic abduction method largely corrects. Additionally, referring to Table 6 in [4], our effectiveness is on par with theirs.
>
> > **“For MorphoMNIST, there is no reason to use composition...”**
>
> We agree that the method suggested by the reviewer is valid. Instead, we follow [5,6], which uses ground truth morphological measurement functions provided by the MorphoMNIST library to measure the intensity, thickness, and slant of the counterfactual digits directly. Both methods are valid since they use ground truth mechanisms to evaluate counterfactuals. For MorphoMNIST, we only used pseudo-oracles for attributes which don’t have a ground truth mechanism, i.e. digit class ($d$).
>
> > **“Proposed method does not beat HVAE by a noticeable margin…”**
>
> Table 1 shows that for many settings, i.e. spatial and semantic mechanisms with $\omega=3, p_\varnothing=0.1$, our model exceeds HVAE effectiveness. Notably, the spatial and semantic mechanisms improve digit class accuracy by ~4%. When considering effectiveness across all parents holistically, while the target intervention may be successful in the baselines, the image's defining characteristic (digit class) is not preserved.
>
> As requested, we also train VCI (which outperforms DiffSCM) for the scenario in Fig 2a. Again, all interventions cause large drops in digit class effectiveness:
>
> | Interv. | MAPE(t) | MAPE(i) | MAPE(s) | Acc(d) | Rev. ($L_1$) |
> |-|-|-|-|-|-|
> | do(s) | 3.08e-2 | 6.52e-3 | 9.07e-2 | 90.04 | 3.19e-2 |
> | do(d) | 2.63e-2 | 6.32e-3 | 9.12e-2 | 94.62 | 1.98e-2 |
> | do(t) | 3.08e-2 | 6.26e-3 | 9.13e-2 | 92.97 | 2.10e-2 |
> | do(i) | 6.99e-2 | 2.61e-2 | 3.97e-1 | 82.52 | 3.75e-2 |
>
> with Comp. ($L_1$) = 1.31e-2, with full table [here](https://imgur.com/a/BYuLIgx).
>
> We also compare against VCI trained in a simpler setting using only $i$ and $t$ from the dataset generated with Fig 2a, and also notice large margins of improvement:
>
> |  | MAPE(t) | MAPE(i) | Rev. ($L_1$) | MAPE(t) | MAPE(i) | Rev. ($L_1$) | Comp. ($L_1$) |
> |-|-|-|-|-|-|-|-|
> | VCI | 5.09e-2 | 9.49e-3 | 1.17e-2 | 9.58e-2 | 5.89e-2 | 2.74e-2 | 6.65e-3 |
> | Spatial | 3.45e-2 | 6.31e-3 | 3.19e-2 | 5.52e-2 | 1.07e-2 | 3.53e-2 | 9.26e-4 |
>
> where the first three cols are for do(t) and the next three are for do(i), with the full table provided [here](https://imgur.com/a/IojOgnj).
>
> Note that given our choice of metrics differs from those in VCI, the margins of improvement are not comparable across works.
>
> Additionally, we run VCI on CelebAHQ without center cropping, such that effectiveness can be evaluated with our pseudo-oracles, and we notice a large drop in the counterfactual fidelity:
>
> |  | F1(g) | F1(s) |
> |-|-|-|
> | do(g) | 3.39 | 97.84 |
> | do(s) | 95.58 | 33.81 |
>
> with examples [here](https://imgur.com/a/DRhkuUY).
>
> We will include DDCM[8] in our related work. DDCM, which uses text-based LDMs, is challenging to compare fairly with our method due to model size, dataset requirements, and prompt engineering. Their sampling method in Alg 1 in [8] complements our work and naturally extends our mechanisms alongside other sampling strategies[7,9]. This is outside the scope of this paper.
>
> We appreciate the reviewer’s detailed feedback and believe these clarifications and additional experiments demonstrate that our work offers an improved and valuable perspective on diffusion-based counterfactual generation.
>
> [1] https://arxiv.org/abs/2212.12570
>
> [2] https://arxiv.org/abs/2410.12730
>
> [3] https://www.jmlr.org/papers/volume23/21-0080/21-0080.pdf
>
> [4] https://arxiv.org/abs/2303.01274
>
> [5] https://arxiv.org/abs/2306.15764
>
> [6] https://github.com/biomedia-mira/causal-gen/blob/e0e4e22f8ad972b9d3b7dd662fa77d9d7c845078/notebooks/eval_example.ipynb
>
> [7] https://arxiv.org/abs/2206.00927
>
> [8] https://arxiv.org/pdf/2312.04965
>
> [9] https://arxiv.org/abs/2406.08070

---

> > ### Comment · Reviewer_kmep · 2025-04-03
> >
> > I thank the authors for the response and I am very impressed with the effort you put in the updated experiments, so I have raised the score by 1. However, I still have a big problem with the metrics you used for morphoMNIST that I think you ignored. Besides, I still have concern that comparison to prior diffusion models in image counterfactual / image editing are not shown. Therefore, I'm going to give the authors a tractable objective: if you solve one of these two issues, I'm going to further raise the score to a 3. If you solve both, I'll raise it to a 4.
> >
> > ---
> >
> > **Issue 1**: morphoMNIST metric
> >
> > I'm not asking for ground truth measurement for the observed factors, I understand you already used those for effectiveness. What I'm saying is using composition (MSE between reconstruction and original image) and reversibility (MSE between cycle reconstruction and original image) over counterfactual error (MSE between counterfactual construction and true counterfactual image) do not make sense when true counterfactual image is available, which is the case for morphoMNIST. The counterfactual error should be used instead of composition and reversibility -- that is a direct estimate of exogenous noise abducation in counterfactual generations.
> >
> > I have some doubt regarding your VCI experiments as the results shown in [2] (Figure 12) do not exhibit any drop in digit class fidelity, especially on intensity which is arguably the easiest intervention. Yet you show a large drop of digit class fidelity specifically on intensity. I find that hard to believe. I'm suspicious if there is something wrong with your classifier. Regardless, you can generate the counterfactual truth with morphoMNIST and if you just show the counterfactual error on the full image along with effectiveness as I suggested, it will be super clear which method is the best. It is a comprehensive evaluation of both exogenous noise and class fidelity. If you only evaluate effectiveness / class fidelity, then yes, conditional VAE-based models such as conditional diffusion models are always going to have the advantage -- interventional inference is what the conditional VAE objective is set out to do.
> >
> > ---
> >
> > **Issue 2**: Baseline model
> >
> > While I believe in the authors that VCI outperforms DiffSCM, I think the latter baseline is more essential to this paper because I think it is very important to show comparison to prior diffusion model in image counterfactual / image editing as an alternative proposal of diffusion model in image counterfactual. Readers would very much want to know in what aspects did your proposed novelties make a difference compared to prior diffusion work. DDCM's main algorithm (Algorithm 1) has nothing to do with text or prompt engineering, but I will give you a pass as I do acknowledge the potential heavy workload in adaptation. While I don't believe your model can beat DDCM, I fully believe your model can beat DiffSCM pretty easily because I don't think DiffSCM was a coherent effort. However, DiffSCM was kind of a landmark work of applying diffusion model to counterfactual modeling and it would be nice to show the comparison such that your paper would be more self-contained. After all, both yours and DiffSCM are non-theoretical papers, so regardless of how much better your high-level idea sounds compared to DiffSCM, empirical evidence is probably the only compelling evidence.
> >
> > ---
> >
> > **Note**: I confirmed with AC that this year's ICML does not allow more than 2 rounds of interactions during the discussion phase, instead, reviewers can update their comments to add additional information. Therefore, I have posted my response to the Authors' **Reply Rebuttal Comment** below in the **Update** paragraph of my original review.

---

> > > ### Author Response · Authors · 2025-04-08
> > >
> > > We thank the reviewer for increasing their score and for providing very valuable, concrete suggestions for further improving our paper. We have now added the requested comparisons and experiments which we hope address the remaining issues.
> > >
> > > ---
> > > **Issue 1**
> > >
> > > As requested, we have computed MSE/MAE between predictions and the "true" counterfactual for two Morpho-MNIST SCMs: (i, t)-only and (i, t, s, d). While the results confirm the competitiveness of our methods, we believe this is not the most meaningful method of evaluation. Pixel-wise metrics are fundamentally flawed when the target is not unique - there are multiple equally plausible counterfactuals that accurately obey interventions which may exhibit spatial ambiguity (such as rotations or translations). Even small spatial differences may induce large changes in pixel-wise metrics despite being unrelated to effectiveness. These metrics are also sensitive to sharpness and local artefacts. A poor but blurry counterfactual may yield low MSE [1,2,8]. Concerns about MSE for counterfactual error have been raised in the VCI reviews by reviewer QZf3 (see link: [https://openreview.net/forum?id=oeDcgVC7Xh&noteId=dkGjENbOX](https://openreview.net/forum?id=oeDcgVC7Xh&noteId=is9vzvhbQQ)).
> > >
> > > MSE/MAE remain meaningful for composition and reversibility because the target is uniquely defined (ie. the observed image). We would also like to stress that the soundness of counterfactuals cannot be assessed by any metric alone, so it is not a question of choosing one over the other. Hence, we report composition and reversibility in addition to effectiveness measured on intervened variables, following the common evaluation strategies in [3,4,5].  Please also refer to the theoretical results in [6,7] which support our evaluation approach.
> > >
> > > Regarding the classifier, upon double-checking, we are confident that it works correctly (we provide implementation and performance in App. D2), and we use the same classifier to evaluate all models which show good performance. The drop in digit class fidelity for VCI is related to the model performing poorly for low-intensity interventions with generated images being almost empty. Please see do(i) visuals generated by the VCI codebase during training at 50 and 280 epochs: https://imgur.com/a/DVDLstb. We attribute this to instability during training, with frequent loss spikes, across all VCI mechanisms we trained: https://imgur.com/a/L31aody.
> > >
> > > **(i,t)-mechanism**
> > > | | do(t) - $L_1 (10^{-2}) / L_2 (10^{-3})$ | do(i) - $L_1 (10^{-2}) / L_2 (10^{-3})$ |
> > > |-|-|-|
> > > | VCI | 1.19/2.53 | 1.43/3.74 |
> > > | Semantic ($p_\varnothing=0.1, \omega = 1$) | 1.89/4.29 | 1.28/3.33 |
> > >
> > > **(i,t,s,d)-mechanism**
> > > | | do(t) - $L_1 (10^{-2}) / L_2 (10^{-3})$ | do(i) - $L_1 (10^{-2}) / L_2 (10^{-3})$ | do(s) - $L_1 (10^{-2}) / L_2 (10^{-2})$ |
> > > |-|-|-|-|
> > > | VCI | 1.73/4.44 | 3.83/2.22 | 3.11/1.23 |
> > > | Spatial ($p_\varnothing=0.5, \omega = 1.5$) | 1.70/4.85 | 1.04/2.42 | 3.31/1.39 |
> > > | Semantic ($p_\varnothing=0.1, \omega = 1.5$) | 1.65/4.35 | 1.49/3.45 | 3.10/1.26 |
> > >
> > > ---
> > >
> > > **Issue 2**
> > >
> > > We would like to thank the reviewer for insisting on the comparison to DiffSCM. We agree this comparison adds value. As suggested, we have now added this comparison by taking DiffSCM’s original source code (which is limited to digit interventions), ensuring that its training hyperparameters follow those in their App E. table 2, and generate counterfactuals with the range of guidance scales provided in their Fig 4. We then trained our digit-conditional spatial and semantic mechanisms to compare with DiffSCM (and VCI). As correctly predicted by the reviewer, our methods outperform DiffSCM.
> > >
> > > | | Comp $L_1$ | Acc % | Rev $L_1$ |
> > > |-|-|-|-|
> > > | VCI | 2.05e-2 | 92.48 | 6.71e-2 |
> > > | DiffSCM ($\omega = 3$) | 8.20e-3 | 17.02 | 2.62e-2 |
> > > | Spatial ($p_\varnothing = 0.1, \omega = 1.5$) | 1.23e-2 | 99.63 | 5.12e-2 |
> > > | Semantic ($p_\varnothing = 0.1, \omega = 1.5$) | 6.83e-3 | 97.46 | 5.05e-2 |
> > >
> > > ---
> > >
> > > [1] Zhang et al. "The unreasonable effectiveness of deep features as a perceptual metric." CVPR 2018
> > >
> > > [2] Wang & Bovik, "Mean squared error: Love it or leave it?" IEEE 2009 - Fig 2
> > >
> > > [3] Hao et al. "Natural Counterfactuals With Necessary Backtracking." Neurips 2024
> > >
> > > [4] Ribeiro et al. “High Fidelity Image Counterfactuals with Probabilistic Causal Models.” ICML 2023.
> > >
> > > [5] Melistas et al. “Benchmarking Counterfactual Image Generation.” NeurIPS 2024 Track on Datasets and Benchmarks
> > >
> > > [6] Monteiro et al. MEASURING AXIOMATIC SOUNDNESS OF COUNTERFACTUAL IMAGE MODELS. ICLR 2023
> > >
> > > [7] Halpern, Axiomatizing Causal Reasoning. JAIR 2000 - Sec 3
> > >
> > > [8] Wang et al, "Image quality assessment: from error visibility to structural similarity." IEEE 2004

---

### Official Review · Reviewer_wWq6 · 2025-03-14

**Overall Recommendation:** 3

**Summary:**

This paper explores diffusion models for counterfactual image generation by incorporating semantic abduction to enhance high-level semantic identity preservation causal consistency. The authors propose a structural causal model (SCM)-based framework that integrates diffusion models for counterfactual reasoning, leveraging spatial, semantic, and dynamic abduction mechanisms. The paper introduces amortized anti-causal guidance to improve intervention fidelity and evaluates the approach using counterfactual soundness metrics across datasets, including Morpho-MNIST, CelebA-HQ, and EMBED.

## update after rebuttal
Thank the authors for their rebuttal. Some of my concerns have been addressed. The experimental section could be further improved by offering a more comprehensive comparison with strong baseline models. Additionally, it would have been beneficial to see results on a broader range of real-world datasets, which would better highlight the practical value of the approach to the vision community. I maintain my original score of weak accept.

**Claims And Evidence:**

Most claims are well-supported with experimental results and theoretical justification.
However, claims related to scalability and complexity may meed further validation.

**Essential References Not Discussed:**

N/A

**Experimental Designs Or Analyses:**

The overall experimental setup is well-designed, including multiple real datasets and quantitative metrics.
Further improvement could consider including large-scale and more complicated datasets, and include some human perceptual evaluation.

**Methods And Evaluation Criteria:**

The proposed methods and evaluation criteria are well-aligned with the problem, but the benchmark datasets are relatively simple, and additional evaluations on more complex, real-world datasets would strengthen the work.

**Other Comments Or Suggestions:**

n/a

**Other Strengths And Weaknesses:**

The approach has promising applications in medical imaging and causal reasoning tasks.

**Questions For Authors:**

- What is the computational cost of the method and have you considered efficiency optimizations to make diffusion-based counterfactual generation more practical?

- How does your method compare to causal disentanglement approaches in generative modeling?

**Relation To Broader Scientific Literature:**

The key contributions of this paper build on prior work in counterfactual generation, causal inference, and diffusion models, integrating these areas in a novel way.

**Theoretical Claims:**

The paper presents theoretical justifications for integrating semantic abduction and counterfactual trajectory alignment (CTA) within diffusion models, particularly in the context of structural causal models (SCMs). The high-level formulation appears sound.

---

> ### Author Rebuttal · Authors · 2025-03-31
>
> We thank the reviewer for the positive comments regarding our framework’s “promising applications in medical imaging and causal reasoning tasks” with “well-designed” experiments and our integration of existing concepts in a “novel way”.
>
> > **Scalability and Computational Costs**
>
> Like most diffusion-based approaches, our method incurs high computational costs at generation time. We acknowledge that techniques such as distillation [1], consistency models [2], and higher-order solvers for sampling [3] can improve efficiency; however, these optimisations are beyond the scope of our current work. We focus on exploring the trade-offs involved in using diffusion models for counterfactual generation, thereby setting the stage for future work addressing efficiency concerns.
>
> Notably, our approach to dynamic semantic abduction via CTA leverages causally conditioned diffusion models to improve identity preservation with a single-step optimisation. This contrasts with methods that rely on less robust, image-specific test-time optimisations - such as expensive multi-step optimisations or heuristics about self and cross-attention maps - often requiring extensive fine-tuning for each image [4, 5, 6, 7]. Our single-step method simplifies and enables rapid testing on larger datasets with a simple model selection process (Fig 5). Dynamic semantic abduction via CTA improves identity preservation of backgrounds, hairstyles, and skin colours (see Fig 3c and App H - Fig 16).
>
> Furthermore, using the latent diffusion model paradigm, our approach can be readily scaled to higher-dimensional images [8]. Our work performs comparably to existing variational methods like HVAEs [9] and VDVAEs [10], which can be challenging to scale to deeper architectures for modelling complex images due to the need to compute and match posteriors [11, 12]. In contrast, the diffusion paradigm uses a fixed posterior, offering a more computationally scalable alternative. We can provide further exposition about this in sec 3.1 for clarity.
>
> > **Comparison to Causal Disentanglement Approaches in Generative Modelling**
>
> In response to reviewer kmep, we provide additional results on VCI [13], whose results improve upon DEAR [14], a popular causal disentanglement generative modelling approach. Our results show that our methods improve effectiveness in the scenario in Fig. 2c, and in a simpler causal modelling problem involving only thickness and intensity, whilst reversibility decreases as guidance scale increases due to trajectory deviation, as described in Sec 3.3. We also focus on methods where an SCM can be assumed instead of methods that infer the assumed causal graph but may struggle with scaling to larger parent sets [16]. We intend to provide further exposition about the challenges associated with defining an SCM in our limitations, and future work could explore incorporating causal discovery frameworks akin to [17].
>
> When compared to CausalDiffAE [17], we condition on $c_{sem} = (z_{sem}, pa)$, whereas they condition solely on $z_{sem}$, which also encodes the parent SCM in the style of [16]. Unfortunately, we could not reproduce the results in CausalDiffAE for the main paper or during the rebuttal period; instead, we have included another baseline from the aforementioned work on VCI [13] in response to reviewer kmep. On inspection of Fig 2a in CausalDiffAE, we believe that their framework may perform poor abduction when all parents are unobserved, as in many real-world scenarios, given their poor preservation of digit style.
>
> [1] https://arxiv.org/abs/2303.01469
>
> [2] https://arxiv.org/abs/2202.00512
>
> [3] https://arxiv.org/abs/2206.00927
>
> [4] https://arxiv.org/abs/2211.09794
>
> [5] https://arxiv.org/abs/2309.15664
>
> [6] https://arxiv.org/abs/2405.01496v1
>
> [7] https://arxiv.org/abs/2212.12570
>
> [8] https://arxiv.org/abs/2112.10752
>
> [9] https://arxiv.org/abs/2306.15764
>
> [10] https://arxiv.org/abs/2303.01274
>
> [11] https://arxiv.org/abs/2401.06281
>
> [12] https://arxiv.org/abs/2208.11970
>
> [13] https://arxiv.org/abs/2410.12730
>
> [14] https://www.jmlr.org/papers/volume23/21-0080/21-0080.pdf
>
> [15] https://arxiv.org/abs/2004.08697
>
> [16] https://arxiv.org/abs/2004.08697
>
> [17] https://arxiv.org/abs/2404.17735

---

### Official Review · Reviewer_vbtG · 2025-03-16

**Overall Recommendation:** 3

**Summary:**

This paper studies the image counterfactual generation problem using diffusion models. Specifically, the authors propose a suite of deep causal mechanisms, spatial mechanism, semantic mechanism, and anti-causal mechanism, for tractable counterfactual generation with respect to composition, reversibility, and effectiveness. The authors also propose a dynamic classifier-free training approach to study the trade-off between composition and effectiveness of generated counterfactuals. Experiments on three datasets (one synthetic and two real-world) are performed to evaluate the soundness of generated image counterfactuals.


## Update After Rebuttal
I appreciate the authors taking the time to address my questions and concerns. I believe this work has some limitations, however, after the rebuttal period, I believe the authors did a good job responding to all the reviewers' concerns. I will keep my score at 3 and lean towards **acceptance** of this paper. However, it would be good if the authors made their contribution more clear in the writing and distinguish it clearly from different approaches (DCM, CausalDiffAE, DiffSCM). Since there are only a few diffusion counterfactual baselines and the authors have included some results from DiffSCM (in response to Reviewer kmep), I am satisfied with comparisons. Ideally there would be comparisons with more diffusion counterfactual baselines. I will note that DiffSCM is not completely related to the paradigm proposed here. Although DiffSCM proposes a general formulation for arbitrary causal graphs, it only applies to classifier-based guidance (e.g., image and its label) in practice. Therefore, I do not think this baseline is all that much related to methods such as this paper, DCM, and CausalDiffAE which explicitly focus on modeling the causal structure.

**Claims And Evidence:**

Yes

**Essential References Not Discussed:**

N/A

**Experimental Designs Or Analyses:**

Yes, I checked the soundness of the experimental design and analysis. Specifically, the use of the theoretically grounded compositionality, reversibility, and effectiveness metrics have been shown to be good axiomatic metrics to evaluate the soundness of generated counterfactuals. Furthermore, I checked the visual quality of generated counterfactuals for all datasets. Overall, the evaluation is sound, extensive, and holistic.

**Methods And Evaluation Criteria:**

Yes

**Other Comments Or Suggestions:**

N/A

**Other Strengths And Weaknesses:**

## Strengths
- The paper is written exceptionally well with clear intuitions and formulations contextualizing diffusion-based counterfactual generation with respect to composition, reversibility, and effectiveness, which are fundamental notions in counterfactual inference.
- The experiments are extensive and show that the proposed abduction mechanisms are quite effective in counterfactual generation on several different counterfactual soundness metrics. Furthermore, the authors experiment on facial image data and a breast cancer image dataset, which further underscores the application of the method in real-world domains.

## Weaknesses

- The proposed mechanisms seem to be generalizations of existing methods. For instance, the spatial mechanism is essentially a generalized version of DCM (Chao et al.), the semantic mechanism is a generalized version of DiffAE (Preechakul et al.) and CausalDiffAE (Komanduri et al.), and the anti-causal mechanism is a generalization of DiffSCM (Sanchez et al.). Although the comprehensive formulation is important, the main contribution is not clear.
- The authors take a causal interpretation of classifier-free guidance to show that the masking probability of the semantic conditioning acts as a trade-off between composition and effectiveness. However, Komanduri et al also take a very similar causal interpretation of classifier-free guidance. There does not seem to be a substantial difference between the two interpretations.

**Questions For Authors:**

- Could the authors provide clarifications on the differences between the proposed mechanisms and related work as pointed out in the weakness section?
- Could the authors explain the difference between their causal interpretation of classifier-free guidance in Eq (16) and the one proposed by CausalDiffAE (Komanduri et al)?
- What is the motivation behind learning the $\phi$ token for dynamic classifier free guidance? Traditionally, one would just use a zero-mask an jointly train a conditional and unconditional diffusion model.
- How does this framework translate to text-to-image diffusion models? For existing pretrained models, such as Stable Diffusion, it can be unrealistic to retrain them using this sort of paradigm. How would one perform high-fidelity counterfactual image generation using these pretrained models?

Komanduri et al. Causal Diffusion Autoencoders: Toward Counterfactual Generation via Diffusion Probabilistic Models. ECAI 2024.

**Relation To Broader Scientific Literature:**

This paper provides a general framework and formalization for counterfactual generation using diffusion models. Specifically, the paper extends the general structure of Pawlowski et al and Ribeiro et al to provide a suite of deep causal mechanisms for tractable counterfactual generation. The causal mechanisms serve as formal generalizations of previous diffusion counterfactual methods, namely DCM (Chao et al), CausalDiffAE (Komanduri et al), and DiffSCM (Sanchez et al).


Pawlowski et al. Deep Structural Causal Models for Tractable Counterfactual Inference. NeurIPS 2020.

Ribeiro et al. High Fidelity Image Counterfactuals with Probabilistic Causal Models. ICML 2023.

Chao et al. Modeling Causal Mechanisms with Diffusion Models for Interventional and Counterfactual Queries. TMLR 2024.

Komanduri et al. Causal Diffusion Autoencoders: Toward Counterfactual Generation via Diffusion Probabilistic Models. ECAI 2024.

Sanchez et al. Diffusion Causal Models for Counterfactual Estimation. CLeaR 2022.

**Theoretical Claims:**

N/A

---

> ### Author Rebuttal · Authors · 2025-03-31
>
> We thank the reviewer for the positive feedback and are pleased that the paper was found to be “written exceptionally well with clear intuitions” and that the experiments “are extensive and show that the proposed abduction mechanisms are quite effective”. Below we address weaknesses and questions:
>
> > **“Although the comprehensive formulation is important, the main contribution is not clear”...**
>
> Concretely, our work distinguishes itself from existing approaches by:
> - Spatial Abduction: We use causally conditioned DDIM inversion rather than the unconditional forward process [1,2,3], such that spatial noise can help preserve image structure [4]. We go beyond DiffSCM, by incorporating multiple causal conditions (discrete or continuous), and DCM, by demonstrating our mechanisms on complex vision datasets. Also, we adopt a new SCM for MorphoMNIST, including a challenging relationship between slant and digit.
> - Semantic Abduction: Our semantic mechanisms condition diffusion models on semantic encoding and causal conditions from the parent’s SCM, whereas CausalDiffAE [2] incorporates [5] into diffusion models, widely acknowledged to be sensitive to a large number of parents and susceptible to unstable training [6,7], diminishing the benefits of the stable training objective of diffusion models. DiffAE does not provide a method for amortised guidance. Our analysis goes significantly beyond CausalDiffAE and DiffAE, exploring how our amortised guidance procedure trades off effectiveness for identity preservation.
> - Dynamic Abduction: We introduce a novel, general, dynamic semantic abduction framework, which we implement via CTA, that learns image-specific guidance tokens during the abduction step in counterfactual inference. These tokens enhance the retention of attributes that cannot be trivially measured and included as causal conditions via the SCM, such as background, hairstyle, and skin colour (Fig 3c, App H - Fig 16).
>
> Our formulation provides a unifying perspective between image editing using pre-trained diffusion models and counterfactual image generation methods. As such, we set the stage for future work on model identifiability and incorporating techniques from LDM-based methods to improve dynamic abduction, which has been a primarily empirical field. Additionally, by carefully considering which variables should be regarded as exogenous or observed within our formalism, we can evaluate counterfactuals via metrics grounded by intuitive soundness axioms and more formally explain tradeoffs between intervention-faithfulness (effectiveness) and identity preservation (composition) when using diffusion models for counterfactual inference or image editing.
>
> > **Difference between their causal interpretation of classifier-free guidance in Eq (16) and the one proposed by CausalDiffAE**
>
> We condition on $c_{sem} = (z_{sem}, pa)$, whereas they condition solely on $z_{sem}$ which also encodes the parent SCM in the style of [5].
>
> > **What is the motivation behind learning the ϕ token for dynamic classifier-free guidance?**
>
> As stated in sec 3.3, we train our guided mechanisms in the standard way [8]. We only use CTA to perform dynamic semantic abduction during counterfactual inference. Given that the CTA optimisation is performed w.r.t composition, the guidance tokens can learn additional semantic information not captured by $z_{sem}$ or $pa$, thereby improving the preservation of hair colour, skin tone and backgrounds in counterfactuals (Fig 3c, App H - Fig 16).
>
> > **How does this framework translate to text-to-image diffusion models?**
>
> Our general dynamic abduction procedure (eq 21-22) is analogous to many test-time optimisations for LDM-based image editing [4, 9, 10, 11], with the fundamental difference that fine-grained causal control can be challenging via text conditioning alone [12]. For an LDM, dynamic abduction using the spatial mechanism with amortised guidance (eq 19) can be formulated trivially within our framework. Natural extensions of our work involve replacing CTA with the test-time optimisation techniques cited above. Further, we demonstrate that shorter DDIM strides with single-step guidance token updates at each timestep improve identity preservation over semantic abduction. In contrast, many LDM-based methods opt for longer DDIM strides with computationally expensive multi-step guidance token optimisations.
>
> [1] https://arxiv.org/abs/2210.11841
>
> [2] https://arxiv.org/abs/2202.10166
>
> [3] https://arxiv.org/abs/2404.17735
>
> [4] https://arxiv.org/abs/2211.09794
>
> [5] https://arxiv.org/abs/2004.08697
>
> [6] https://ieeexplore.ieee.org/document/10021114
>
> [7] https://arxiv.org/abs/2411.19556
>
> [8] https://arxiv.org/abs/2207.12598
>
> [9] https://arxiv.org/abs/2309.15664
>
> [10] https://arxiv.org/abs/2405.01496v1
>
> [11] https://arxiv.org/abs/2403.02981
>
> [12] https://arxiv.org/abs/2212.12570
>
> [13] https://arxiv.org/abs/2410.12730

---

> > ### Comment · Reviewer_vbtG · 2025-04-04
> >
> > I appreciate the authors providing clarifications to my questions and concerns. Currently, the core contribution of this work is still a bit unclear to me. I understand the suite of mechanisms provided. However, they really seem to be generalizations of existing approaches (e.g., DCM [1], CausalDiffAE [2], DiffSCM [3]). For the semantic abduction mechanism, conditioning on $c_{sem}=(z_{sem}, pa)$ does not explicitly encode any causal mechanisms. For instance, if we consider DeepSCM [4], we can see that the causal relationship between variables is explicitly modeled via a normalizing flow. In my view, it seems that $c_{sem}$ should be a compact representation that explicitly embeds causal relationships (similar to the role $z_{causal}$ plays in CausalDiffAE).
> >
> > The following is my rationale for keeping my score at a 3. I believe this paper is quite well written and provides a generalized framework for diffusion-based counterfactual modeling. Furthermore, I believe the experiments are extensive and serve as interesting case studies into the utility of each of the mechanisms with robust metric evaluations. That said, the reason I am not inclined to give anything higher than a 3 is because I believe the fundamental contribution is weak compared to existing diffusion-based counterfactual generation work (e.g., DCM [1], CausalDiffAE [2], DiffSCM [3]) as well as prior counterfactual generation work in the VAE setting (e.g., DeepSCM [4], CausalHVAE [5]).
> >
> >
> > [1] Chao et al. Modeling Causal Mechanisms with Diffusion Models for Interventional and Counterfactual Queries. TMLR 2024.
> >
> > [2] Komanduri et al. Causal diffusion autoencoders: towards counterfactual generation via diffusion probabilistic models. ECAI 2024.
> >
> > [3] Sanchez et al. Diffusion Causal Models for Counterfactual Estimation. CLeAR 2022.
> >
> > [4] Pawlowski et al. Deep Structural Causal Models for Tractable Counterfactual Inference. NeurIPS 2020.
> >
> > [5] Ribeiro et al. High Fidelity Image Counterfactuals with Probabilistic Causal Models. ICML 2023.

---

> > > ### Author Response · Authors · 2025-04-08
> > >
> > > We thank the reviewer for taking the time to read our response.
> > >
> > > Regarding contributions, we would like to highlight that the existing diffusion-based models performed poorly in our experiments, and we believe that our work marks a fundamental advancement in generating high-quality counterfactuals with diffusion models. We tackle key limitations of previous work: CausalDiffAE [2] clearly does not preserve identity in their Fig 2a, DiffSCM [3] is limited to a single discrete parent [3] and admits poor effectiveness and image fidelity in their Fig 3b (see our 2nd response to reviewer kemp), and DCM [1] shows no evidence of being directly applicable to complex imaging datasets. We believe the poor identity preservation in CausalDiffAE is due to $z_{causal}$ encoding both semantic identity and causal parents, so an intervention affects identity. By keeping $z$ separate from $pa$ we are able to better preserve identity with our semantic mechanism. The strength of our method over previous diffusion-based work is further confirmed by the additional experiments we carried out as part of the rebuttal. We also outperform existing VAE-based models [4,5,6].
> > >
> > > In terms of methodological contribution, please note that in addition to our spatial and semantic methods which you mention, we also present dynamic abduction which is an entirely novel method that has not been discussed previously which is highly effective for preserving backgrounds, hairstyles and skin colours (Fig 3c + App H - Fig 16). Current VAE-based methods struggle with these.
> > >
> > > [1] Chao et al. Modeling Causal Mechanisms with Diffusion Models for Interventional and Counterfactual Queries. TMLR 2024.
> > >
> > > [2] Komanduri et al. Causal diffusion autoencoders: towards counterfactual generation via diffusion probabilistic models. ECAI 2024. - Fig 2a
> > >
> > > [3] Sanchez et al. Diffusion Causal Models for Counterfactual Estimation. CLeAR 2022. - Sec. 3.4 & 5
> > >
> > > [4] Pawlowski et al. Deep Structural Causal Models for Tractable Counterfactual Inference. NeurIPS 2020.
> > >
> > > [5] Ribeiro et al. High Fidelity Image Counterfactuals with Probabilistic Causal Models. ICML 2023.
> > >
> > > [6] Wu et al. Counterfactual Generative Modeling with Variational Causal Inference. ICLR 2025

---

### Official Review · Reviewer_v9YA · 2025-03-17

**Overall Recommendation:** 4

**Summary:**

The paper “Diffusion Counterfactual Generation with Semantic Abduction” explores the use of diffusion models for counterfactual image generation, a task that requires maintaining identity, visual fidelity, and causal consistency. The authors argue that while diffusion models have achieved state-of-the-art synthesis quality, they lack structured semantic control for counterfactual reasoning. To address this, the paper introduces a new framework that integrates diffusion models with structural causal models (SCMs), leveraging semantic abduction to improve identity preservation in generated counterfactuals. The proposed approach introduces mechanisms such as spatial, semantic, and dynamic abduction to refine causal interventions while maintaining perceptual consistency. The study evaluates its methods across synthetic and real-world datasets, including face images and medical imagery, demonstrating improvements over existing generative approaches like VAEs and GANs. However, challenges such as computational efficiency, dataset limitations, and reliance on human-defined causal graphs are acknowledged. The work presents an important step towards using generative models for more controlled, interpretable, and robust counterfactual reasoning.

**Claims And Evidence:**

The paper’s core claims—that diffusion models generate high-quality counterfactuals and semantic abduction improves identity preservation—are well-supported through experiments and ablation studies. However, it overstates the ease of defining causal structures, as real-world causal relationships are often ambiguous. Additionally, while results on faces and medical images are promising, generalization to more complex, diverse datasets is not fully explored. More testing on real-world counterfactual scenarios would strengthen its conclusions.

**Essential References Not Discussed:**

The related works are well discussed

**Experimental Designs Or Analyses:**

Yes, I reviewed the experimental design and analysis validity, the experimental design is well-structured and supports key claims, but the lack of large-scale, diverse datasets and human validation weakens its generalizability. More real-world testing and sensitivity analysis on causal assumptions would improve robustness.

**Methods And Evaluation Criteria:**

The proposed method—integrating diffusion models with causal reasoning—is a logical and promising approach for counterfactual image generation, as diffusion models excel at high-quality synthesis while causal inference ensures meaningful interventions. The use of semantic abduction to refine counterfactual edits also makes sense, as it helps preserve identity and realism.

However, the evaluation criteria have some limitations. While the paper uses perceptual similarity metrics, identity preservation scores, and qualitative comparisons, it lacks human evaluation to assess the realism and plausibility of generated counterfactuals. Additionally, the datasets (faces and medical images), though relevant, are relatively small and lack diversity, making it unclear how well the method generalizes to more complex, real-world counterfactual reasoning tasks. More robust benchmarking on larger and varied datasets would improve the evaluation.

**Other Comments Or Suggestions:**

Please refer to previous section

**Other Strengths And Weaknesses:**

Please refer to previous section

**Questions For Authors:**

Please refer to previous section

**Relation To Broader Scientific Literature:**

This paper builds on diffusion models for high-quality image synthesis (Ho et al., 2020) and extends them to counterfactual generation, an area previously dominated by GANs and VAEs (Goyal et al., 2020). Unlike prior methods, it integrates Structural Causal Models (SCMs) to enforce causal consistency, bridging generative modeling with causal inference. The introduction of semantic abduction refines counterfactual edits, improving identity preservation and realism beyond standard interventions.

**Theoretical Claims:**

The paper does not contain formal, theorem-based proofs that require rigorous correctness checks. Instead, it presents a theoretical framework for integrating diffusion models with causal inference, specifically through semantic abduction.

---

> ### Author Rebuttal · Authors · 2025-03-31
>
> We thank the reviewer for the constructive feedback and for recognising the significance of our work. We're encouraged by reviewers acknowledging our framework as "an important step towards using generative models for more controlled, interpretable, and robust counterfactual reasoning". We also appreciate the positive remarks regarding our integration of diffusion models with counterfactual inference, the novelty of semantic abduction, and the strength of our experimental design and ablation studies. Additionally, we want to highlight further the novelty, generality, and effectiveness of our dynamic abduction framework, which also offers opportunities to integrate alternative test-time optimisations from LDM-based image editing into structured counterfactual inference beyond CTA.
>
> > **"It lacks human evaluation to assess the realism and plausibility of generated counterfactuals."**
>
> We agree that incorporating human evaluation to assess the plausibility and realism of generated counterfactuals would add value to our analysis. In particular, expert assessment by radiologists for EMBED and experienced annotators for CelebAHQ could provide additional insights beyond our chosen perceptual metrics. However, such user studies are time-consuming and challenging to set up and not feasible as part of the rebuttal. We acknowledge this limitation and plan to include it as a key direction for future work in our discussion section.
>
> > **"However, it overstates the ease of defining causal structures, as real-world causal relationships are often ambiguous."**
>
> We recognise that our manuscript may have understated the challenges involved in specifying causal structures, especially in real-world applications. We will expand the limitations section to acknowledge the ambiguity in causal discovery, the assumptions required to define SCMs, and the fact that our results are contingent on the accuracy of our SCM. Note that if assumptions in an SCM are shown to be incorrect, the transparency of our framework means that developers can redesign their SCM, and either retrain our mechanisms under new assumptions or perform more faithful simulated interventions.
>
> For MorphoMNIST, we leverage the known data-generating process within our DSCM directly for the parents of the image. In the CelebAHQ experiments, we adopt a simplified SCM involving attributes such as smiling and eyeglasses with mild label noise, which can be reliably predicted from observed images using pseudo-oracles, consistent with prior work (see Appendix F, [1]). For the EMBED dataset, our SCM is derived from clinical insights presented in [2] and validated by radiologists.
> We also note that discovering causal structure is an active and open research area [3,4], and future work could focus on jointly learning causal structure and counterfactual generation, taking inspiration from the theoretical discussion in Appendix C in [5].
>
> > **"Generalization to more complex, diverse datasets is not fully explored. More testing on real-world counterfactual scenarios would strengthen its conclusions."**
>
> We appreciate the reviewer’s concern regarding generalisability. While we agree that broader testing is important, we would like to highlight that the EMBED artefact removal task addresses a clinically important and underexplored problem in a real-world medical imaging scenario. The EMBED variant proposed by [2] contains many patients with significant variations in tissue density and anatomical structure. We believe this makes EMBED a suitably complex testbed for evaluating the robustness of our counterfactual generation methods, directly addressing a real-world challenge of breast cancer detection in screening mammography. Compared to many existing works, we felt that including this real-world medical imaging application sets our work apart from the many ML papers that primarily test on readily available benchmarks.
>
> Furthermore, we have obtained improved results on EMBED since the initial submission, achieving over 90% accuracy in triangular artefact removal. These updated findings will be included in the paper's final version to support our semantic mechanisms' efficacy further.
>
> [1] https://arxiv.org/abs/2303.01274
>
> [2] https://arxiv.org/abs/2410.03809
>
> [3] https://arxiv.org/abs/2004.08697
>
> [4] https://arxiv.org/abs/2307.05704
>
> [5] https://arxiv.org/abs/2404.17735

---

### Decision · Program_Chairs · 2025-05-01

**Decision:**

Accept (poster)

**Comment:**

All reviewers generally appreciated the work, felt the paper was well-written, and believed the experiments were convincing.

The main concerns were about comparison to diffusion-based methods but the author rebuttal and discussion was able to resolve these concerns to reasonable satisfaction.

Please make sure to add the new experiments and metrics along with anything else discussed throughout the author-reviewer discussion period. These new results and explanations were critical for the final paper's quality.